# Stability-penalty-adaptive
# follow-the-regularized-leader:
# Sparsity, game-dependency, and best-of-both-worlds

**Taira Tsuchiya**\*
The University of Tokyo
tsuchiya@mist.i.u-tokyo.ac.jp

**Shinji Ito**
NEC Corporation / RIKEN
i-shinji@nec.com

**Junya Honda**
Kyoto University / RIKEN
honda@i.kyoto-u.ac.jp

## Abstract

Adaptivity to the difficulties of a problem is a key property in sequential decision-making problems to broaden the applicability of algorithms. Follow-the-regularized-leader (FTRL) has recently emerged as one of the most promising approaches for obtaining various types of adaptivity in bandit problems. Aiming to further generalize this adaptivity, we develop a generic adaptive learning rate, called *stability-penalty-adaptive (SPA) learning rate* for FTRL. This learning rate yields a regret bound jointly depending on stability and penalty of the algorithm, into which the regret of FTRL is typically decomposed. With this result, we establish several algorithms with three types of adaptivity: *sparsity*, *game-dependency*, and *best-of-both-worlds* (BOBW). Despite the fact that sparsity appears frequently in real problems, existing sparse multi-armed bandit algorithms with $k$-arms assume that the sparsity level $s \leq k$ is known in advance, which is often not the case in real-world scenarios. To address this issue, we first establish $s$-agnostic algorithms with regret bounds of $\widetilde{O}(\sqrt{sT})$ in the adversarial regime for $T$ rounds, which matches the existing lower bound up to a logarithmic factor. Meanwhile, BOBW algorithms aim to achieve a near-optimal regret in both the stochastic and adversarial regimes. Leveraging the SPA learning rate and the technique for $s$-agnostic algorithms combined with a new analysis to bound the variation in FTRL output in response to changes in a regularizer, we establish the first BOBW algorithm with a sparsity-dependent bound. Additionally, we explore partial monitoring and demonstrate that the proposed SPA learning rate framework allows us to achieve a game-dependent bound and the BOBW simultaneously.

## 1 Introduction

This study considers the multi-armed bandits (MAB) and partial monitoring (PM). In the MAB problem, the learner selects one of $k$ arms, and the adversary simultaneously determines the loss of each arm, $\ell_t = (\ell_{t1}, \ldots, \ell_{tk})^\top$ in $[0, 1]^k$ or $[-1, 1]^k$. After that, the learner observes only the loss for the chosen arm. The learner's goal is to minimize the regret, which is the difference between the learner's total loss and the total loss of an optimal arm fixed in hindsight. PM is a generalization of MAB, and the learner observes feedback symbols instead of the losses.

---

\*This work was done when the author was with Kyoto University and RIKEN.

37th Conference on Neural Information Processing Systems (NeurIPS 2023).

One of the most promising frameworks for MABs and PM is follow-the-regularized-leader (FTRL) [5, 12], which determines the arm selection probability at each round by minimizing the sum of the cumulative (estimated) losses so far plus a convex regularizer. Note that the well-known Exp3 algorithm developed in [5] is equivalent to FTRL with the (negative) Shannon entropy regularizer. FTRL is also known to perform well for the classic expert problem [17] and reinforcement learning [52]. Furthermore, when the problem is "benign", it is known that FTRL can exploit the underlying structure to adaptively improve its performance. Typical examples of such adaptive improvements are (i) *data-dependent bounds* and (ii) *best-of-both-worlds* (BOBW).

Data-dependent bounds have been investigated to enhance the adaptivity of algorithms to a given structure of losses in the *adversarial regime*, where feedback (*e.g.,* losses in MABs) is decided in an arbitrary manner. There are various examples of data-dependent bounds, and this study considers *sparsity-dependent bounds* and *game-dependent bounds*.

A sparsity-dependent bound is an important example of data-dependent bounds, as sparsity frequently appears in real-world problems. For example, in online advertisement allocation, it is often the case that only a fraction of the ads is clicked. Although there are some studies for sparse MABs [10, 27, 51], all of them assume that (an upper bound of) sparsity level $s \geq \|\ell_t\|_0 = |\{i \in [k] \colon \ell_{ti} \neq 0\}|$ is known beforehand, which in many practical scenarios does not hold.

The concept of a game-dependent bound was recently introduced by Lattimore and Szepesvári [32] to derive a regret upper bound that depends on the game the learner is facing. As the authors suggest, one of the motivations for the game-dependent bound is that previous PM algorithms are "quite conservative and not practical for normal problems". For example, whereas the Bernoulli MAB is expressed as a PM, algorithms for PM do not always achieve the minimax regret of MAB [5]. The game-dependent bound enables the learner to automatically adapt to the essential difficulty of the game the algorithm is actually facing.

The BOBW algorithm aims to achieve near-optimal regret bounds in stochastic and adversarial regimes, where the feedback is stochastically generated in the stochastic regime. Since we often do not know the underlying regime, it is desirable for an algorithm to *simultaneously* obtain a near-optimal performance both for the stochastic and adversarial regimes. For multi-armed bandits, Bubeck and Slivkins [9] developed the first BOBW algorithm, and Zimmert and Seldin [53] proposed the well-known Tsallis-INF algorithm, which achieves the optimal regret for both regimes. The Tsallis-INF algorithm also achieves favorable regret guarantees in the *adversarial regime with a self-bounding constraint*, which interpolates between the stochastic and adversarial regimes.

To realize the aforementioned adaptivity in FTRL, the *adaptive learning rate* (a.k.a. time-varying learning rate) is one of the most representative approaches. This approach adjusts the learning rate based on previous observations. In the literature, adaptive learning rates have been designed to depend on *stability* or *penalty*, which are components of a regret upper bound of FTRL. The stability term increases if the variation of FTRL outputs in the adjacent rounds is large, and stability-dependent learning rates have been used in a considerable number of algorithms available in the literature, *e.g.,* [32, 37, 38] and references therein. In contrast, the penalty term comes from the strength of the regularization, and recently penalty-dependent learning rates were considered to achieve BOBW guarantees [22, 46]. However, existing stability-dependent (resp. penalty-dependent) learning rates are designed with the worst-case penalty (resp. stability), which could potentially limit the adaptivity and performance of FTRL. (There are numerous studies related to this paper and we include additional related work in Appendix C.)

## 1.1 Contribution of this study

In this paper, in order to further broaden the applicability of FTRL, we establish a generic framework for designing an adaptive learning rate that depends on both the stability and penalty components simultaneously, which we call a *stability-penalty-adaptive (SPA) learning rate* (Definition 2). This enables us to bound the regret approximately by $\widetilde{O}\big(\sqrt{\sum_{t=1}^T z_t h_{t+1}}\big)$ for stability component $(z_t)_t$ and a penalty component $(h_t)_t$, which we call a *SPA regret bound* (Theorem 1). With appropriate selections of $z_t$ and $h_t$, this result yields the three important adaptive bounds mentioned earlier, namely sparsity, game-dependency, and BOBW. In particular, our contributions are as follows (see also Tables 1 and 2):

Table 1: Regret upper bounds with sparsity-dependent bounds in multi-armed bandits. $T$ is the time horizon. $s \leq k$ is the level of sparsity in losses. Let $L_2 = \sum_{t=1}^{T} \|\ell_t\|^2$, and $\|\ell_t\|_0 \leq s$ implies $L_2 = \sum_{t=1}^{T} \|\ell_t\|^2 \leq sT$ since $\|\ell_t\|_\infty \leq 1$. $\Delta_{\min}$ is the minimum suboptimality gap. Adv. and Stoc. are the abbreviations of the adversarial and stochastic regime, respectively.

| Reference | $s$-agnostic? | Range of $\ell_{ti}$ | Regime | Regret bound |
|---|---|---|---|---|
| Kwon and Perchet [27] | – | $[0,1]$ | Adv. | $\Omega(\sqrt{sT})$ |
| Kwon and Perchet [27] | No | $[0,1]$ | Adv. | $2\sqrt{e}\sqrt{sT\log(k/s)}$ |
| **Ours (Sec. 5.1.1, Cor. 2)** | Yes | $[0,1]$ | Adv. | $2\sqrt{2}\sqrt{L_2\log k} + O((kT\log k)^{1/3})$ |
| Bubeck et al. [10] | No | $[-1,1]$ | Adv. | $10\sqrt{L_2\log k} + 20k\log T$ |
| **Ours (Sec. 5.1.2, Cor. 3)** | Yes | $[-1,1]$ | Adv. | $4\sqrt{2}\sqrt{L_2\log k} + 2k\log T$ |
| **Ours (Sec. 5.2, Thm. 4)** | Yes | $[-1,1]$ | Adv. | $4\sqrt{L_2\log k \log T} + O(k\log T)$ |
| | | | Stoc. | $O(s\log(T)\log(kT)/\Delta_{\min})$ |

Table 2: Regret bounds for non-degenerate local PM games. $V_t$, $V_t'$, and $\bar{V}'$ are game-dependent quantities satisfying $V_t \leq V_t' \leq \bar{V}$ (see Section 6 and Appendix B for definitions). $H(q_t)$ is the Shannon entropy for FTRL output $q_t$.

| Reference | Game-dependent? | BOBW? | Order of regret bound |
|---|---|---|---|
| Many existing studies on PM | No | No | – |
| Lattimore and Szepesvári [32] | Yes | No | $\sqrt{\sum_{t=1}^{T} V_t \log k}$ |
| Tsuchiya et al. [46] | No (only game-class-dependent) | Yes | $\sqrt{\bar{V}\sum_{t=1}^{T} H(q_{t+1})}$ |
| **Ours (Sec. 6, Cor. 5)** | Yes | Yes | $\sqrt{\sum_{t=1}^{T} V_t' H(q_{t+1})\log T}$ |

- (Section 5.1) We initially provide new algorithms for sparse MABs as preliminaries for establishing a BOBW algorithm with a sparsity-dependent bound. In Section 5.1.1, we propose a novel estimator of the sparsity level, which is linked to a stability component and induces $L_2 = \sum_{t=1}^{T} \|\ell_t\|^2 \leq sT$. We demonstrate that a learning rate using this estimator with the Shannon entropy regularizer and $\widetilde{\Theta}((kT)^{-2/3})$ uniform exploration immediately results in an $O(\sqrt{L_2\log k})$ regret bound for $\ell_t \in [0,1]^k$. In Section 5.1.2, we investigate possibly negative losses $\ell_t \in [-1,1]^k$. We employ the time-invariant log-barrier proposed in [10] to control the stability term. This allows us to achieve an $O(\sqrt{L_2\log k})$ regret bound for losses in $[-1,1]^k$ even *without* the $\widetilde{\Theta}((kT)^{-2/3})$ uniform exploration. This is a key component for developing the BOBW guarantee that we discuss next. Note that Section 5.1 serves as preliminary findings for the subsequent section.

- (Section 5.2) We establish a BOBW algorithm with a sparsity-dependent bound. In order to achieve this goal, we make another major technical development: we analyze the variation in the FTRL output when the regularizer changes (Lemma 7), which holds thanks to the time-invariant log-barrier and may be of independent interest. This analysis is necessary since we use a time-varying learning rate to obtain a BOBW guarantee, whereas Bubeck et al. [10] uses a constant learning rate. This technical development successfully allows us to achieve the goal (Theorem 4) in combination with the SPA learning rate developed in Section 4 and a technique for exploiting sparsity in Section 5.1.2.

- (Section 6) We show that the SPA learning rate established in Section 4 can also be used to achieve a game-dependent bound and a BOBW guarantee simultaneously, which further highlights the usefulness of the SPA learning rate.

## 2 Setup

This section introduces the preliminaries of this study. Sections 2.1 and 2.2 formulate the MAB and PM problems, respectively, and Section 2.3 defines regimes considered in this paper.

**Notation** Let $\|x\|$, $\|x\|_1$, and $\|x\|_\infty$ be the Euclidian, $\ell_1$-, and $\ell_\infty$-norms for a vector $x$, respectively. Let $\|x\|_0$ be the number of non-zero elements for a vector $x$. Let $\mathcal{P}_k = \{p \in [0,1]^k : \|p\|_1 = 1\}$ be the $(k-1)$-dimensional probability simplex. A vector $e_i \in \{0,1\}^k$ is the $i$-th standard basis of $\mathbb{R}^k$, and $\mathbf{1}$ is the all-one vector. Let $D_\Phi$ be the *Bregman divergence* induced by differentiable convex function $\Phi$, *i.e.,* $D_\Phi(p,q) = \Phi(p) - \Phi(q) - \langle \nabla\Phi(q), p - q\rangle$. Table 3 in Appendix A summarizes the notation used in this paper.

## 2.1 Multi-armed bandits

In MAB with $k$-arms, at each round $t \in [T] := \{1, 2, \ldots, T\}$, the environment determines the loss vector $\ell_t = (\ell_{t1}, \ell_{t2}, \ldots, \ell_{tk})^\top$ in $[0,1]^k$ or $[-1,1]^k$, and the learner simultaneously chooses an arm $A_t \in [k]$ without knowing $\ell_t$. After that, the learner observes only the loss $\ell_{tA_t}$ for the chosen arm. The performance of the learner is evaluated by the regret $\mathrm{Reg}_T$, which is the difference between the cumulative loss of the learner and of the single optimal arm, that is, $a^* = \arg\min_{a\in[k]} \mathbb{E}\big[\sum_{t=1}^T \ell_{ta}\big]$ and $\mathrm{Reg}_T = \mathbb{E}\big[\sum_{t=1}^T (\ell_{tA_t} - \ell_{ta^*})\big]$, where the expectation is taken with respect to the internal randomness of the algorithm and the randomness of the loss vectors $(\ell_t)_{t=1}^T$.

## 2.2 Partial monitoring

**Formulation** A PM game $\mathcal{G} = (\mathcal{L}, \Phi)$ with $k$-actions and $d$-outcomes is defined by a pair of a loss matrix $\mathcal{L} \in [0,1]^{k\times d}$ and feedback matrix $\Phi \in \Sigma^{k\times d}$, where $\Sigma$ is a set of feedback symbols. The game is played in a sequential manner by a learner and an opponent across $T$ rounds. The learner begins the game with knowledge of $\mathcal{L}$ and $\Phi$. For every round $t \in [T]$, the opponent selects an outcome $x_t \in [d]$, and the learner simultaneously chooses an action $A_t \in [k]$. Then the learner suffers an unobserved loss $\mathcal{L}_{A_t x_t}$ and receives only a feedback symbol $\sigma_t = \Phi_{A_t x_t}$, where $\mathcal{L}_{ax}$ is the $(a,x)$-th element of $\mathcal{L}$. The learner's performance in the game is evaluated by the regret $\mathrm{Reg}_T$ as in the MAB case: $a^* = \arg\min_{a\in[k]} \mathbb{E}\big[\sum_{t=1}^T \mathcal{L}_{ax_t}\big]$ and $\mathrm{Reg}_T = \mathbb{E}\big[\sum_{t=1}^T (\mathcal{L}_{A_t x_t} - \mathcal{L}_{a^* x_t})\big] = \mathbb{E}\big[\sum_{t=1}^T \langle \ell_{A_t} - \ell_{a^*}, e_{x_t}\rangle\big]$, where $\ell_a \in \mathbb{R}^d$ is the $a$-th row of $\mathcal{L}$.

**Several concepts in PM** Let $m \le |\Sigma|$ be the maximum number of distinct symbols in a single row of $\Phi \in \Sigma^{k\times d}$. Different actions $a$ and $b$ are duplicate if $\ell_a = \ell_b$. We can decompose possible distributions of $d$ outcomes in $\mathcal{P}_d$ based on the loss matrix. For every action $a \in [k]$, cell $\mathcal{C}_a = \{u \in \mathcal{P}_d : \max_{b\in[k]}(\ell_a - \ell_b)^\top u \le 0\}$ is the set of probability vectors in $\mathcal{P}_d$ for which action $a$ is optimal. Each cell is a closed convex polytope.

Define $\dim(\mathcal{C}_a)$ as the dimension of the affine hull of $\mathcal{C}_a$. Action $a$ is said to be dominated if $\mathcal{C}_a = \emptyset$. For non-dominated actions, action $a$ is said to be Pareto optimal if $\dim(\mathcal{C}_a) = d - 1$, and degenerate if $\dim(\mathcal{C}_a) < d - 1$. Let $\Pi$ be the set of Pareto optimal actions. Two Pareto optimal actions $a, b \in \Pi$ are called neighbors if $\dim(\mathcal{C}_a \cap \mathcal{C}_b) = d - 2$, which is used to define the difficulty of PM games. A PM game is said to be non-degenerate if it has no degenerate actions. We assume that PM game $\mathcal{G}$ is non-degenerate and contains no duplicate actions.

The difficulty of PM games is characterized by the following observability conditions. Neighbouring actions $a$ and $b$ are locally observable if there exists $w_{ab} : [k] \times \Sigma \to \mathbb{R}$ such that $w_{ab}(c, \sigma) = 0$ for $c \notin \{a, b\}$ and $\sum_{c=1}^k w_{ab}(c, \Phi_{cx}) = \mathcal{L}_{ax} - \mathcal{L}_{bx}$ for all $x \in [d]$. A PM game is locally observable if all neighboring actions are locally observable, and this study considers locally observable games.

**Loss difference estimation** Let $\mathcal{H}$ be the set of all functions from $[k] \times \Sigma$ to $\mathbb{R}^d$. For any locally observable games, there exists $G \in \mathcal{H}$ such that for any $b, c \in \Pi$, $\sum_{a=1}^k (G(a, \Phi_{ax})_b - G(a, \Phi_{ax})_c) = \mathcal{L}_{bx} - \mathcal{L}_{cx}$ for all $x \in [d]$ [32]. For example, we can take $G = G_0$ defined by $G_0(a, \sigma)_b = \sum_{e\in\mathrm{path}_{\mathcal{T}}(b)} w_e(a, \sigma)$ for $a \in \Pi$, where $\mathcal{T}$ is a tree over $\Pi$ induced by neighborhood relations and $\mathrm{path}_{\mathcal{T}}(b)$ is the set of edges from $b \in \Pi$ to an arbitrarily chosen root $c \in \Pi$ on $\mathcal{T}$ [32]. See Appendix C and [31, Chapter 37] for a more detailed explanation and background of PM.

## 2.3 Considered regimes

We consider three regimes on the assumptions for losses in MABs and outcomes in PM. In the *stochastic regime*, a sequence of loss vector $(\ell_t)$ in MAB and that of outcome vector $(x_t)$ in PM follow an unknown distribution $\nu^*$ in an i.i.d. manner. Define the minimum suboptimality gap in $\Delta_{\min} = \min_{a \neq a^*} \Delta_a$ for $\Delta_a = \mathbb{E}_{\ell_t \sim \nu^*}[(\ell_{ta} - \ell_{ta^*})]$ in MAB and $\Delta_a = \mathbb{E}_{x_t \sim \nu^*}[(\ell_a - \ell_{a^*})^\top e_{x_t}]$ in PM. Note that the definitions of $\ell$ in MAB and PM are different.

In contrast, the *adversarial regime* does not assume any stochastic structure for the losses or outcomes, and they can be chosen in an arbitrary manner. In this regime, the environment can choose $\ell_t$ for MAB and $x_t$ for PM depending on the past history until the $(t-1)$-th round, $(A_s)_{s=1}^{t-1}$.

We also consider, the *adversarial regime with a self-bounding constraint* [53], an intermediate regime between the stochastic and adversarial regimes.

**Definition 1.** *Let $\Delta \in [0, 2]^k$ and $C \geq 0$. The environment is in an* adversarial regime with a $(\Delta, C, T)$ *self-bounding constraint if it holds for any algorithm that* $\mathrm{Reg}_T \geq \mathbb{E}\left[\sum_{t=1}^T \Delta_{A_t} - C\right]$.

One can see that the stochastic and adversarial regimes are indeed instances of this regime, and that well-known *stochastic regimes with adversarial corruptions* [36] are also in this regime (see [53] and [46] for definitions in MAB and PM, respectively).

We assume that there exists a unique optimal arm (or action) $a^*$, which was employed by many studies aiming at developing BOBW algorithms [18, 34, 49, 53].

# 3 Preliminaries

This section provides preliminaries for developing and analyzing algorithms. We first introduce FTRL, upon which we develop our algorithms, and then describe the self-bounding technique, which is a common technique for proving a BOBW guarantee.

**Follow-the-regularized-leader** In the FTRL framework, an arm selection probability $p_t \in \mathcal{P}_k$ at round $t$ is given by

$$q_t = \arg\min_{q \in \mathcal{P}_k} \left\langle \sum_{s=1}^{t-1} \widehat{y}_s, \, q \right\rangle + \Phi_t(q) \quad \text{and} \quad p_t = \mathcal{T}_t(q_t), \tag{1}$$

where $\widehat{y}_t \in \mathbb{R}^k$ is an estimator of loss $\ell_t$ at round $t$, $\Phi_t \colon \mathcal{P}_k \to \mathbb{R}$ is a strongly-convex regularizer, and $\mathcal{T}_t \colon \mathcal{P}_k \to \mathcal{P}_k$ is a map from the output of FTRL $q_t$ to an arm selection probability vector $p_t$.

In the analysis of FTRL, it is common to evaluate $\sum_{t=1}^T \langle \widehat{y}_t, p_t - p \rangle = \sum_{t=1}^T \langle \widehat{y}_t, q_t - p \rangle + \sum_{t=1}^T \langle \widehat{y}_t, p_t - q_t \rangle$ for some $p \in \mathcal{P}_k$. It is known (see *e.g.,* [31, Exercise 28.12]) that quantity $\sum_{t=1}^T \langle \widehat{y}_t, q_t - p \rangle$ is bounded from above by

$$\sum_{t=1}^T \underbrace{(\Phi_t(q_{t+1}) - \Phi_{t+1}(q_{t+1}))}_{\text{penalty term}} + \Phi_{T+1}(p) - \Phi_1(q_1) + \sum_{t=1}^T \underbrace{(\langle q_t - q_{t+1}, \widehat{y}_t \rangle - D_{\Phi_t}(q_{t+1}, q_t))}_{\text{stability term}}. \tag{2}$$

We refer to the terms in (2) as a *penalty* and *stability* terms, and to the quantity $\langle \widehat{y}_t, p_t - q_t \rangle$ as a *transformation* term. Note that, though this study focuses on examples in which $\Phi_{T+1}(p)$ is not dominant, this term may be dominant dependent on the choice of regularizers.

**Self-bounding technique** A self-bounding technique is a common method for proving a BOBW guarantee [18, 49, 53]. In the self-bounding technique, we first derive regret upper and lower bounds in terms of a variable dependent on the arm selection probabilities $(p_t)_t$ or the FTRL outputs $(q_t)_t$, and then derive a regret bound by combining the upper and lower bounds. We use a version proposed in [22]. We consider $Q(i)$, $\bar{Q}(i)$, $P(i)$, and $\bar{P}(i)$ for $i \in [k]$ defined by $Q(i) = \sum_{t=1}^T (1 - q_{ti})$, $\bar{Q}(i) = \mathbb{E}[Q(i)]$, $P(i) = \sum_{t=1}^T (1 - p_{ti})$, and $\bar{P}(i) = \mathbb{E}[P(i)]$. Note that $\bar{Q}(i), \bar{P}(i) \in [0, T]$ for any $i \in [k]$. In terms of $\bar{Q}(i)$ or $\bar{P}(i)$, we can obtain the lower bound of the regret for the adversarial regime with a self-bounding constraint as follows:

**Lemma 1** ([46, Lemma 4]). *In the adversarial regime with a self-bounding constraint (Definition 1), if there exists $c' \in (0,1]$ such that $p_{ti} \geq c' q_{ti}$ for all $t \in [T]$ and $i \in [k]$, then $\mathsf{Reg}_T \geq \Delta_{\min} \bar{P}(a^*) - C \geq c' \Delta_{\min} \bar{Q}(a^*) - C$.*

It is known that the sums of the entropy $H(\cdot)$ of $(p_t)_t$ is bounded by $P(i)$ as follows:

**Lemma 2** ([22, Lemma 4]). *Let $(q_t)_{t=1}^T$ be any sequence of probability vectors and define $Q(i) = \sum_{t=1}^T (1 - q_{ti})$. Then for any $i \in [k]$, $\sum_{t=1}^T H(q_t) \leq Q(i) \log(ekT/Q(i))$.*

Based on Lemmas 1 and 2, it suffices to show $\mathsf{Reg}_T \lesssim \mathbb{E}\left[\sqrt{\sum_{t=1}^T H(q_t) \operatorname{polylog}(T)}\right]$ to prove a BOBW guarantee in MAB. This is because, for the adversarial regime, using $H(q_t) \leq \log k$ immediately implies a $\widetilde{O}(\sqrt{T})$ bound, and for the stochastic regime, using Lemmas 1 and 2 roughly bounds the regret as $\mathsf{Reg}_T = 2\mathsf{Reg}_T - \mathsf{Reg}_T \lesssim \sqrt{\bar{Q}(a^*) \operatorname{polylog}(T)} - \Delta_{\min} \bar{Q}(a^*) \lesssim \operatorname{polylog}(T)/\Delta_{\min}$.

## 4 Stability-penalty-adaptive (SPA) learning rate and regret bound

This section proposes a new adaptive learning rate, which yields a regret upper bound dependent on both the stability component $z_t$ and penalty component $h_t$ for various choices of $z_t$ and $h_t$. When we use a learning rate $\eta_t$, the analysis of FTRL boils down to the evaluation of

$$\widehat{\mathsf{Reg}}_T^{\mathsf{SP}} = \sum_{t=1}^T \left(\frac{1}{\eta_{t+1}} - \frac{1}{\eta_t}\right) h_{t+1} + \lambda \sum_{t=1}^T \eta_t z_t \quad \text{for some} \quad \lambda > 0. \tag{3}$$

In particular, when we use the Exp3 algorithm, $h_t$ is the Shannon entropy of the FTRL output at round $t$. This can be confirmed by checking the existing studies (*e.g.,* [22, 46]) or the proofs in Appendices F, G, H.2, and I. To favorably bound $\widehat{\mathsf{Reg}}_T^{\mathsf{SP}}$, we develop a new learning rate framework, which we call the jointly stability- and penalty-adaptive learning rate, or the *stability-penalty-adaptive (SPA) learning rate* for short:

**Definition 2** (Stability-penalty-adaptive learning rate). *Let $((h_t, z_t, \bar{z}_t))_{t=1}^T$ be non-negative reals such that $h_1 \geq h_t$ for all $t \in [T]$, $(\bar{z}_t h_1 + \sum_{s=1}^t z_s h_{s+1})_{t=1}^T$ is non-decreasing, and $\bar{z}_t h_1 \geq z_t h_{t+1}$ for all $t \in [T]$. Let $c_1, c_2 > 0$. Then, a sequence of $(\eta_t)_{t=1}^T$ is a SPA learning rate if it has a form of*

$$\eta_t = \frac{1}{\beta_t}, \quad \beta_1 > 0, \quad \text{and} \quad \beta_{t+1} = \beta_t + \frac{c_1 z_t}{\sqrt{c_2 + \bar{z}_t h_1 + \sum_{s=1}^{t-1} z_s h_{s+1}}}. \tag{4}$$

**Remark.** To the best of our knowledge, this is the first learning rate that depends on both the stability and penalty components. Note that when we set the penalties to their worst-case value, that is, $h_t = h_1$ for all $t \in [T]$ (recalling $h_t \leq h_1$), the SPA learning rate in (4) becomes equivalent to the standard type of the learning rate, which depends only on the stability and has the form of $\beta_t = 1/\eta_t \simeq \frac{c_1}{\sqrt{h_1}} \sqrt{\bar{z}_1 + \sum_{s=1}^{t-1} z_s}$. On the other hand, when we set the stabilities to be their worst-case value, that is, $z \geq \max_{t \in [T]} z_t$, the SPA learning rate in (4) corresponds to the learning rate dependent only on the penalty in [22, 46].

Using learning rate $(\eta_t)$ in (4), we can bound $\widehat{\mathsf{Reg}}_T^{\mathsf{SP}}$ as follows.

**Theorem 1** (Stability-penalty-adaptive regret bound). *Let $(\eta_t)_{t=1}^T$ be a SPA learning rate in Definition 2. Then $\widehat{\mathsf{Reg}}_T^{\mathsf{SP}}$ in (3) is bounded as follows:*

(I) *If $((h_t, z_t, \bar{z}_t))_{t=1}^T$ in $(\eta_t)$ satisfies $\frac{\sqrt{c_2 + \bar{z}_t h_1}}{c_1}(\beta_1 + \beta_t) \geq \varepsilon + z_t$ for all $t \in [T]$ for some $\varepsilon > 0$ (stability condition* (S1)*), then*

$$\widehat{\mathsf{Reg}}_T^{\mathsf{SP}} \leq 2\left(c_1 + \frac{\lambda}{c_1} \log\left(1 + \sum_{u=1}^T \frac{z_u}{\varepsilon}\right)\right) \sqrt{c_2 + \bar{z}_T h_1 + \sum_{t=1}^T z_t h_{t+1}}. \tag{5}$$

(II) *If $h_t = h_1$ for all $t \in [T]$, $c_2 = 0$, and $((h_t, z_t, \bar{z}_t))_{t=1}^T$ in $(\eta_t)$ satisfies $\beta_t \geq \frac{ac_1}{\sqrt{h_1}} \sqrt{\sum_{s=1}^t z_s}$ for some $a > 0$ (stability condition* (S2)*), then $\widehat{\mathsf{Reg}}_T^{\mathsf{SP}} \leq 2\left(c_1 + \frac{\lambda}{ac_1}\right) \sqrt{h_1 \sum_{t=1}^T z_t}$.*

The proof of Theorem 1 can be found in Appendix D. In Part (I) of Theorem 1, we can see that $\widehat{\mathrm{Reg}}_T^{\mathsf{SP}}$ is bounded by $\sqrt{\sum_{t=1}^T z_t h_{t+1}}$, which will enable us to obtain BOBW and data-dependent bounds simultaneously. Note that $\widehat{\mathrm{Reg}}_T^{\mathsf{SP}}$, which is the component of the regret, often becomes dominant in particular when we use the Shanon entropy regularizer. Thus, checking the stability condition (S1) and applying Theorem 1 to bound $\widehat{\mathrm{Reg}}_T^{\mathsf{SP}}$ almost complete the regret analysis. In Section 6, we will see that applying Theorem 1 immediately provides a BOBW bound with a game-dependent bound for PM. In contrast, when deriving BOBW with a sparsity-dependent bound for MAB in Section 5, we will develop additional techniques and conduct further analysis, for example, to satisfy the stability condition (S1), making use of the time-invariant log-barrier regularizer.

## 5 Sparsity-dependent bounds in multi-armed bandits

This section establishes several sparsity-dependent bounds. We use the FTRL framework in (1) with the inverse weighted estimator $\widehat{y}_t \in \mathbb{R}^k$ given by $\widehat{y}_{ti} = \ell_{ti} \mathbb{1}[A_t = i]/p_{ti}$. This estimator is common in the literature and is useful for its unbiasedness, *i.e.*, $\mathbb{E}_{A_t \sim p_t}[\widehat{y}_t \mid p_t] = \ell_t$. We first propose algorithms that achieve sparsity-dependent bounds using stability-dependent learning rates in Section 5.1 as preliminaries for the subsequent section. Following that, in Section 5.2, we establish a BOBW algorithm with a sparsity-dependent bound based on the SPA learning rate. More specific steps are summarized as follows.

- Section 5.1.1 discusses the case of $\ell_t \in [0, 1]^k$ and shows that appropriately choosing $z_t$ in the SPA learning rate (4) with the Shannon entropy regularizer and $\widetilde{\Theta}((kT)^{-2/3})$ uniform exploration achieves a $O(\sqrt{L_2 \log k})$ regret for $\ell_t \in [0, 1]^k$ without knowing $L_2$.

- Section 5.1.2 considers the case of $\ell_t \in [-1, 1]^k$, which is known to be more challenging than $\ell_t \in [0, 1]^k$. We show that the time-invariant log-barrier enables us to choose a "tighter" $z_t$ in (4), which removes the uniform exploration used in Section 5.1.1. This not only results in the bound of $O(\sqrt{L_2 \log k})$ for $\ell_t \in [-1, 1]^k$ but also becomes one of the key properties to achieve BOBW.

- Section 5.2 presents a BOBW algorithm with a sparsity-dependent bound using the technique developed in Section 5.1 and Theorem 1. While Theorem 1 itself is a strong tool leading directly to the result for PM (Section 6), its application does not lead to the desired bounds. In particular, in this setting the $\widetilde{O}\big(\sqrt{\sum_{t=1}^T z_t h_{t+1}}\big)$ term derived through Theorem 1 does not immediately imply a BOBW guarantee with a sparsity-dependent bound. To solve this problem, we develop a novel technique to analyze *the variation in FTRL outputs $q_t$ in response to the change in a regularizer (Lemma 7)*, and prove a BOBW bound with a sparsity-dependent bound of $O(\sqrt{L_2 \log k \log T})$.

### 5.1 Parameter-agnostic sparsity-dependent bounds

This section establishes $s$-agnostic algorithms to achieve sparsity-dependent bounds for the adversarial regime, which are preliminaries for Section 5.2.

#### 5.1.1 $L_2$-agnostic algorithm with $O(\sqrt{L_2 \log k})$ bound for $\ell_t \in [0, 1]^k$

Here, we use $p_t = \mathcal{T}_t(q_t)$ for $\mathcal{T}_t(q) = (1 - \gamma)q + \frac{\gamma}{k}\mathbf{1}$ and $\gamma = \frac{k^{1/3}(\log k)^{1/3}}{T^{2/3}}$ and assume $\gamma \in [0, 1/2]$ (this holds when $T \geq \sqrt{8k \log k}$), which implies $2p_t \geq q_t$. We use the Shannon entropy regularizer $\Phi_t(p) = -\frac{1}{\eta_t} H(p) = \frac{1}{\eta_t}\psi^{\mathsf{nS}}(p) = \frac{1}{\eta_t}\sum_{i=1}^k p_i \log p_i$ with learning rate $\eta_t = 1/\beta_t$ and

$$\beta_1 = \frac{2c_1}{\sqrt{h_1}}\sqrt{\frac{k}{\gamma}}, \quad \beta_{t+1} = \beta_t + \frac{c_1 \omega_t}{\sqrt{\log k}\sqrt{\frac{k}{\gamma} + \sum_{s=1}^{t-1}\omega_s}} \quad \text{for} \quad \omega_t := \frac{\ell_{tA_t}^2}{p_{tA_t}}, \tag{6}$$

which corresponds to the learning rate in Definition 2 with $h_t \leftarrow H(q_1) = \log k$, $z_t \leftarrow \omega_t$, $\bar{z}_t \leftarrow k/\gamma$, and $c_2 \leftarrow 0$. The uniform exploration is used to satisfy stability condition (S2) in Theorem 1, the amount of which is determined by balancing the regret coming from the uniform exploration and stability condition (S2). Theorem 1 immediately gives the following bound.

**Corollary 2.** *When $T \geq \sqrt{8k \log k}$, the above algorithm with $c_1 = 1/\sqrt{2}$ achieves* $\mathsf{Reg}_T \leq 2\sqrt{2}\sqrt{L_2 \log k} + (2\sqrt{2} + 1)(kT \log k)^{1/3}$ *without knowing $L_2$. In particular, when $T \geq 7k^2/s^3$,* $\mathsf{Reg}_T \leq (4\sqrt{2} + 1)\sqrt{sT \log k}$.

The proof is given in Appendix F. The most striking feature of the algorithm is its $L_2$ (or $s$)-agnostic property. This is essentially made possible by the learning rate using the data-dependent quantity $\omega_t$ in (6), which satisfies $\mathbb{E}\left[\sqrt{\sum_{t=1}^{T} \omega_t}\right] \leq \sqrt{\sum_{t=1}^{T} \mathbb{E}[\omega_t]} = \sqrt{\sum_{t=1}^{T} \sum_{i=1}^{k} \ell_{ti}^2} = \sqrt{L_2}$. The leading constant of the bound is better than the existing bounds, as shown in Table 1, despite its agnostic property. Note that while the first-order bound in [49] implies the above sparsity-dependent bound when $\ell_t \in [0, 1]^k$, this does not hold when $\ell_t \in [-1, 1]$, which we investigate in the following (see Appendix K for details). We will see in Section 5.1.2 that this assumption can be totally removed by adding a time-invariant log-barrier regularization.

### 5.1.2 $L_2$-agnostic algorithm with $O(\sqrt{L_2 \log k} + k \log T)$ bound for $\ell_t \in [-1, 1]^k$

Here, we consider the case of $\ell_t \in [-1, 1]^k$. It is worth noting that the negative loss cannot be handled by simply shifting the loss since it removes the sparsity from the losses $(\ell_t)$; see [10, 27] and Appendix K for further details. We directly use the output $q_t$ as $p_t$, that is, $p_t = q_t$. We use the hybrid regularizer consisting of the negative Shannon entropy and the log-barrier function, $\Phi_t(p) = \frac{1}{\eta_t}\psi^{\mathsf{nS}}(p) + 2\psi^{\mathsf{LB}}(p)$, where $\psi^{\mathsf{LB}}(p) = \sum_{i=1}^{k} \log(1/x_i)$. We use learning rate $\eta_t = 1/\beta_t$ and

$$\beta_1 = \frac{c_1^2}{8h_1}, \quad \beta_{t+1} = \beta_t + \frac{c_1 \nu_t}{\sqrt{\log k}\sqrt{\nu_t + \sum_{s=1}^{t-1} \nu_s}} \quad \text{for} \quad \nu_t := \omega_t \min\left\{1, \frac{p_{tA_t}}{2\eta_t}\right\}, \quad (7)$$

where $\omega_t$ is defined in (6). Learning rate (7) corresponds to that in Definition 2 with $h_t \leftarrow H(q_1) = \log k$, $z_t \leftarrow \nu_t$, $\bar{z}_t \leftarrow \nu_t$, and $c_2 \leftarrow 0$. We then have the following bound:

**Corollary 3.** *If we run the above algorithm with $c_1 = \sqrt{2}$,* $\mathsf{Reg}_T \leq 4\sqrt{2}\sqrt{L_2 \log k} + 2k \log T + k + 1/4$, *which implies that* $\mathsf{Reg}_T \leq 4\sqrt{2}\sqrt{sT \log k} + 2k \log T + k + 1/4$.

The proof is given in Appendix G. Corollary 3 removes the assumption of $T \geq 7k^2/s^3$ in Corollary 2, and it also improves the leading constant of the regret in [10]. Note that one can prove a bound of the same order, but with a worse leading constant, by setting $\beta_1 \geq 15k$ and combining the analysis similar to that in Section 5.1.1 and the stability bound in [10]. We successfully remove the assumption of $T \geq 7k^2/s^3$ thanks to the following lemma, which serves as one of the key elements in achieving a BOBW guarantee with a sparsity-dependent bound (The proof is given in Appendix G.):

**Lemma 3** (Stability bound for negative losses). *Let $\ell_t \in [-1, 1]^k$ and $\widehat{y}_{ti} = \ell_{ti}\mathbb{1}[A_t = i]/p_{ti}$ be the inverse weighted estimator. Assume that $q_t \leq \delta p_t$ for some $\delta \geq 1$. Then the stability term of FTRL with the hybrid regularizer $\Phi_t = \frac{1}{\eta_t}\psi^{\mathsf{nS}} + 2\delta\,\psi^{\mathsf{LB}}$ is bounded as*

$$\langle q_t - q_{t+1}, \widehat{y}_t \rangle - D_{\Phi_t}(q_{t+1}, q_t) \leq \delta\eta_t \frac{\ell_{tA_t}^2}{p_{tA_t}} \min\left\{1, \frac{p_{tA_t}}{2\eta_t}\right\} = \delta\eta_t\nu_t.$$

**Remark.** We can observe from Lemma 3 that the stability term is bounded in terms of $\nu_t$, and the most important observation is that this $\nu_t$ is bounded by the inverse of the learning rate $1/(2\eta_t) = \beta_t/2$, *i.e.*, $\nu_t \leq \beta_t/2$. This enables us to guarantee the stability condition (S2) in Theorem 1 without needing to mix the $\widetilde{\Theta}((kT)^{-2/3})$ uniform exploration used in Section 5.1.1. Moreover, this will be a key property to prove a BOBW with a sparsity-dependent bound in the next section.

As a minor contribution, by directly bounding the stability component, the RHS of Lemma 3 has a smaller leading constant than the bound obtained by using the bound in [10].

### 5.2 Best-of-both-worlds guarantee with sparsity-dependent bound

Finally, we are ready to establish a BOBW algorithm with a sparsity-dependent bound and derive its regret bound. We use $p_t = \mathcal{T}_t(q_t) = (1 - \gamma)q_t + \frac{\gamma}{k}\mathbf{1}$ with $\gamma = \frac{k}{T}$ (*i.e.*, $\Theta(1/T)$ uniform exploration)

and assume $\gamma \in [0, 1/2]$, which implies $2p_t \geq q_t$ and $\nu_t \leq T$. We use the hybrid regularizer $\Phi_t = \frac{1}{\eta_t}\psi^{\mathsf{nS}} + 4\psi^{\mathsf{LB}}$ with learning rate $\eta_t = 1/\beta_t$ and

$$\beta_1 = 15k, \quad \beta_{t+1} = \beta_t + \frac{c_1 \nu_t}{\sqrt{81c_1^2 + \nu_t h_{t+1} + \sum_{s=1}^{t-1} \nu_s h_{s+1}}} \quad \text{with} \quad h_t = \frac{1}{1 - \frac{k}{T}} H(p_t), \quad (8)$$

where $\nu_t$ is defined in (7). Note that this learning depends on both the stability and penalty components. This corresponds to the SPA learning rate in Definition 2 with $z_t \leftarrow \nu_t$, $\bar{z}_t \leftarrow \nu_t h_{t+1}/h_1$, and $c_2 \leftarrow 81c_1^2$. One can see that stability assumption (S1) in Part (I) of Theorem 1 are satisfied thanks to $\nu_t \leq \beta_t$ (see Appendix H for the proof). We then have the following bounds.

**Theorem 4** (BOBW with sparsity-dependent bound). *Suppose that $T \geq 2k$. Then the above algorithm with $c_1 = \sqrt{2\log(1 + T/\beta_1)}$ (Algorithm 2 in Appendix H) achieves*

$$\mathsf{Reg}_T \leq 4\sqrt{L_2 \log(k) \log(1 + T)} + O(k \log T)$$

*in the adversarial regime,*

$$\mathsf{Reg}_T = O\left( \frac{s \log(T) \log(kT)}{\Delta_{\min}} + \sqrt{\frac{Cs \log(T) \log(kT)}{\Delta_{\min}}} \right)$$

*in the adversarial regime with a $(\Delta, C, T)$ self-bounding constraint, and $\mathsf{Reg}_T = O(\mathbb{E}[\sum_{i=1}^{k} \ell_{ti}^2] \log(T) \log(kT)/\Delta_{\min})$ in the stochastic regime.*

The proof is given in Appendix H. This is the first BOBW bound with the sparsity-dependent ($L_2$-dependent) bound. Note that the bounds in the stochastic regime and the adversarial regime with a self-bounding constraint can also exploit the propoerty of the underlying losses. The bound in the stochastic regime is suboptimal in two respects: its dependence on $\Delta_{\min}$ and $(\log T)^2$. Concurrently, two separate studies improve each suboptimality ([25] for $\Delta_{\min}$ and [13] for $(\log T)^2$), but it is highly uncertain if we can prove a BOBW with *a sparsity-dependent bound* based on their approach, and it is an important future work to investigate this problem.

**Key elements of the proof** In the following, we describe some key elements of the proof of Theorem 4. We need to solve one remaining technical issue. Using Part (I) of Theorem 1, we can show that the regret is roughly bounded by $\mathbb{E}\left[\sqrt{\sum_{t=1}^{T} \nu_t h_{t+1}}\right] \leq \sqrt{\sum_{t=1}^{T} \mathbb{E}[\nu_t h_{t+1}]}$. However, this quantity cannot be straightforwardly bounded since $h_{t+1}$ depends on $\nu_t$.

To address this issue, we analyze the behavior of arm selection probabilities when the regularizer changes. In particular, we first prove in Lemma 7 that $h_{t+1} \leq h_t + k(\beta_{t+1}/\beta_t - 1)h_{t+1}$. This lemma can be proven by a novel analysis evaluating the changes of the FTRL outputs when the learning rate varies (given in Appendices H.1 and H.2) which is not considered and required when we use a time-invariant learning rate (*e.g.,* [10]). Using the last inequality, we have $\sqrt{\sum_{t=1}^{T} \mathbb{E}[\nu_t h_{t+1}]} \lesssim \sqrt{\sum_{t=1}^{T} \mathbb{E}[\nu_t h_t] + k \sum_{t=1}^{T} \mathbb{E}[\nu_t (\beta_{t+1}/\beta_t - 1)h_{t+1}]} \lesssim$
$\sqrt{\sum_{t=1}^{T} \mathbb{E}[\nu_t h_t] + k \sum_{t=1}^{T} \mathbb{E}[(\beta_{t+1} - \beta_t)h_{t+1}]} \lesssim \sqrt{\sum_{t=1}^{T} \mathbb{E}[\nu_t h_t] + k \sum_{t=1}^{T} \mathbb{E}\left[\sqrt{\sum_{t=1}^{T} \nu_t h_{t+1}}\right]} \lesssim$
$\sqrt{\sum_{t=1}^{T} \mathbb{E}[\nu_t h_t]} + k$, which holds thanks again to $\nu_t \leq \beta_t$ and based on the fact that $x \leq \sqrt{a + bx}$ for $a, b, x > 0$ implies $x \lesssim \sqrt{b} + a$ (here we ignore some logarithmic factors). This combined with the self-bounding technique leads to a BOBW guarantee in the stochastic regime.

**Implementation** One may wonder how to compute $\beta_{t+1}$ satisfying (8) since $h_{t+1} = h_{t+1}(\beta_{t+1})$ depends on $\beta_{t+1}$. In fact, this can be computed by defining $F_t : [\beta_t, \beta_t + T] \to \mathbb{R}$ as $F_t(\alpha) = \alpha - \left( \beta_t + c_1 \nu_t / \sqrt{81c_1^2 + \nu_t h_{t+1}(\alpha) + \sum_{s=1}^{t-1} \nu_s h_{s+1}} \right)$ and setting $\beta_{t+1}$ to be a root of $F_t(\alpha) = 0$. Such $\alpha$ can be computed using the bisection method because $F_t$ is continuous (proved in Proposition 2 in Appendix H.3). The detailed discussion can be found in Appendix H.3.

# 6 Best-of-both-worlds with game-dependent bound for partial monitoring

This section discusses the result of a BOBW guarantee with a game-dependent bound for PM. The desired bound is obtained by direct application of the SPA learning rate and Theorem 1, which highlights the usefulness of the SPA learning rate. Due to the space constraints, the background and algorithm are given in Appendix B.

We rely on the Exploration by Optimization (EbO) by Lattimore and Szepesvári [32]. Define

$$
\mathsf{ST}(p, G; q_t, \eta_t) = \max_{x \in [d]} \left[ \frac{(p - q_t)^\top \mathcal{L} e_x}{\eta_t} + \frac{\mathrm{bias}_{q_t}(G; x)}{\eta_t} + \frac{1}{\eta_t^2} \sum_{a=1}^{k} p_a \Psi_{q_t} \left( \frac{\eta_t G(a, \Phi_{ax})}{p_a} \right) \right], \quad (9)
$$

where $\eta_t$ is a learning rate, $\Psi_q(z) = \langle q, \exp(-z) + z - 1 \rangle$ and $\mathrm{bias}_q(G; x) = \langle q, \mathcal{L} e_x - \sum_{a=1}^{k} G(a, \Phi_{ax}) \rangle + \max_{c \in \Pi} (\sum_{a=1}^{k} G(a, \Phi_{ax})_c - \mathcal{L}_{cx})$ is the bias of the estimator. Note that the first and third terms in (9) correspond to the stability and transformation terms. Then in EbO we choose $(p_t, G_t) \in \mathcal{P}_k \times \mathcal{H}$ by

$$
(p_t, G_t) = \operatorname*{arg\,min}_{p \in \mathcal{P}'_k(q_t), G \in \mathcal{H}} \mathsf{ST}(p, G; q_t, \eta_t) \quad \text{for} \quad \mathcal{P}'_k(q) = \{ p \in \mathcal{P}_k : p \geq q/(2k) \}, \quad (10)
$$

where $\mathcal{P}'_k(q)$ is the feasible set proposed in [46]. Let $\mathrm{opt}'_{q_t}(\eta_t)$ be the optimal value of the optimization problem and $V'_t = \max\{0, \mathrm{opt}'_{q_t}(\eta_t)\}$ be its truncation.

**Remark.** It is worth noting that $V'_t = \max\{0, \mathrm{opt}'_{q_t}(\eta_t)\}$ captures game difficulty the learner is facing as discussed in [32]. As discussed above, $\mathsf{ST}(p, G)$ in (9), the objective function that determines $\mathrm{opt}'_{q_t}(\eta_t)$, correspond to the components of the regret upper bound of FTRL. Hence, the smaller $\mathsf{ST}(p, G)$ can become by optimizing $p$ and $G$, the smaller the regret can become, and thus $\mathrm{opt}'_{q_t}(\eta_t)$ captures the game difficulty.

Now we are ready to state our result.

**Corollary 5.** *For any non-degenerate PM games, there exists an algorithm based on a SPA learning rate achieving* $\mathsf{Reg}_T = O\big(\mathbb{E}\big[\sqrt{\sum_{t=1}^{T} V'_t \log(k) \log(1 + T)}\big] + mk^2\sqrt{\log(k)\log(T)}\big)$ *in the adversarial regime, and* $\mathsf{Reg}_T = O(m^2 k^4 \log(T)\log(kT)/\Delta_{\min} + \sqrt{Cm^2 k^4 \log(T)\log(kT)/\Delta_{\min}} + mk^2\sqrt{\log(k)\log(T)})$ *in the adversarial regime with a* $(\Delta, C, T)$ *self-bounding constraint.*

An extended result and the proof are given in Appendices B and I, respectively. Recall that $V'_t = \max\{0, \mathrm{opt}'_{q_t}(\eta_t)\}$ in the bound reflects the difficulty of the game the learner is facing, rather than the worst-case difficulty of the class of the game. The bounds in both regimes are optimal up to logarithmic factors, and further detailed comparisons are given in Table 2 and Appendix B.

# 7 Conclusion and future work

In this paper, we established the stability-penalty-adaptive (SPA) learning rate (Definition 2), which provides the regret upper bound that jointly depends on the stability and penalty components of FTRL (Theorem 1). This learning rate combined with the technique and analysis for bounding stability terms allows us to achieve BOBW and data-dependent bounds (sparsity- and game-dependent bounds) simultaneously in MAB and PM.

There are some remaining questions. First of all, it would be important future direction to apply the SPA learning rate to other online decision-making problems or regularizers. For example, it is as to investigate online learning with feedback graphs [3], in which the Shannon entropy regularizer (or the Tsallis entropy with the exponent larger than $1 - 1/\log k$) is necessary to achieve nearly optimal regret bounds. Another interesting example is to employ the Tsallis entropy as a dominant regularizer in the SPA learning rate, for instance, to improve logarithmic dependences on the regret bounds, while in this paper we only focused on the Shannon entropy. Second, it is an open question whether we can achieve the bound of $O(\sqrt{sT \log(k/s)})$ in the sparsity-dependent bound without knowing the sparsity level $s$. Finally, while we only considered PM games with local observability, investigating if the game-dependent bound with the BOBW guarantee is possible for PM with global observability is important future work.

## Acknowledgments and Disclosure of Funding

TT was supported by JST, ACT-X Grant Number JPMJAX210E, Japan and JSPS, KAKENHI Grant Number JP21J21272, Japan. JH was supported by JSPS, KAKENHI Grant Number JP21K11747, Japan.

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

# A   Notation

Table 3 summarizes the symbols used in this paper.

Table 3: Notation

| Symbol | Meaning |
|---|---|
| $\mathcal{P}_k$ | $(k-1)$-dimensional probability simplex |
| $T \in \mathbb{N}$ | time horizon |
| $k \in \mathbb{N}$ | number of arms (or actions) |
| $A_t \in [k]$ | arm (or action) chosen by learner at round $t$ |
| $s \le k$ | $\max_t \|\ell_t\|_0$, sparsity level of losses |
| $L_2$ | $\sum_{t=1}^{T} \|\ell_t\|^2$ |
| $\mathcal{L} \in [0,1]^{k \times d}$ | loss matrix |
| $\Sigma$ | set of feedback symbols |
| $\Phi \in \Sigma^{k \times d}$ | feedback matrix |
| $d \in \mathbb{N}$ | number of outcomes |
| $m \in \mathbb{N}$ | maximum number of distinct symbols in a single row of $\Phi$ |
| $x_t \in [d]$ | outcome chosen by opponent at round $t$ |
| $q_t \in \mathcal{P}_k$ | output of FTRL at round $t$ |
| $p_t \in \mathcal{P}_k$ | arm selection probability at round $t$ |
| $\Phi_t \colon \mathcal{P}_k \to \mathbb{R}$ | regularizer of FTRL at round $t$ |
| $\eta_t = 1/\beta_t > 0$ | learning rate of FTRL at round $t$ |
| $\psi^{\mathsf{nS}} \colon \mathbb{R}_+^k \to \mathbb{R}$ | $\sum_{i=1}^{k} x_i \log x_i$, negative Shannon entropy |
| $\psi^{\mathsf{LB}} \colon \mathbb{R}_+^k \to \mathbb{R}$ | $\sum_{i=1}^{k} \log(1/x_i)$, log-barrier |
| $\phi^{\mathsf{nS}} \colon \mathbb{R}_+ \to \mathbb{R}$ | $x \log x$ |
| $\phi^{\mathsf{LB}} \colon \mathbb{R}_+ \to \mathbb{R}$ | $\log(1/x)$ |
| $h_t$ | penalty component at round $t$ |
| $z_t$ | stability component at round $t$ |
| $\omega_t$ | stability component $z_t$ introduced in (6) (Section 5.1.1) |
| $\nu_t$ | stability component $z_t$ introduced in (7) (Section 5.1.2) |
| $V_t'$ | stability component $z_t$ introduced in (13) (Appendix B) |
| $C \ge 0$ | corruption level |

# B   Detailed background and algorithm description omitted from Section 6

This section supplements the material that was briefly explained in Section 6. We also consider full information (FI) and MAB as well as non-degenerate locally observable PM (PM-local), and let $\mathcal{M}$ be a such underlying model.

## B.1   Exploration by optimization and its extension

**Exploration-by-optimization**   We first describe the approach of EbO by Lattimore and Szepesvári [32], which is a strong technique to bound the regret in PM with local observability. The key idea behind EbO is to minimize a part of a regret upper bound of the FTRL with the Shannon entropy. Recall that $\mathcal{H}$ is the set of all functions from $[k] \times \Sigma$ to $\mathbb{R}^d$. Then in EbO we consider the choice of $(p_t, G_t) \in \mathcal{P}_k \times \mathcal{H}$ to minimize the sum of the stability and transformation terms for the worst-case outcome given as follows (also defined in (9)):

$$\mathsf{ST}(p, G; q_t, \eta_t) = \max_{x \in [d]} \left[ \frac{(p - q_t)^{\top} \mathcal{L} e_x}{\eta_t} + \frac{\mathrm{bias}_{q_t}(G; x)}{\eta_t} + \frac{1}{\eta_t^2} \sum_{a=1}^{k} p_a \Psi_{q_t} \left( \frac{\eta_t G(a, \Phi_{ax})}{p_a} \right) \right]. \quad (11)$$

Note that the first and third terms in (11) correspond to the stability and transformation terms (divided by the learning rate $\eta_t$), respectively. We define the optimal value of the optimization problem by

**Algorithm 1:** BOBW algorithm with a game-dependent bound for locally observable games

---
**1** **input:** $B$
**2** **for** $t = 1, 2, \ldots$ **do**
**3** $\quad$ Compute $q_t$ using (1).
**4** $\quad$ Solve (10) to determine $V_t' = \max\{0, \mathrm{opt}_{q_t}'(\eta_t)\}$ and the corresponding solution $p_t$ and $G_t$.
**5** $\quad$ Sample $A_t \sim p_t$ and observe feedback $\sigma_t \in \Sigma$.
**6** $\quad$ Compute $\widehat{y}_t = G_t(A_t, \sigma_t)/p_{tA_t}$ and update $\beta_t$ using (13).

---

$\mathrm{opt}_q(\eta) = \min_{(p,G) \in \mathcal{P}_k \times \mathcal{H}} \mathsf{ST}(p, G; q_t, \eta)$ and its truncation at round $t$ by $V_t = \max\{0, \mathrm{opt}_{q_t}(\eta_t)\}$ (appeared in Table 2). Note that this optimization problem is convex and can be solved numerically by using standard solvers [32].

**Extending exploration-by-optimization** While the vanilla EbO is a strong tool to derive a regret bound in PM-local, it only has a guarantee for the adversarial regime. Recall that in the self-bounding technique, we require a lower bound of the regret expressed in terms of $q_t$ (see Lemma 1). However, when we use the vanilla EbO, it may make a certain action selection probability $p_{ta}$ for some action $a$ become extremely small even when the output of FTRL $q_{ta}$ is far from zero [32], which makes it impossible for us to use the self-bounding technique.

To solve this problem, the vanilla EbO was recently extended so that it is applicable to the stochastic regime (and the adversarial regime with a self-bounding constraint) for PM-local [46]. We define $\mathcal{P}_k'(q, \mathcal{M})$ for a class of games $\mathcal{M}$, which is the extended version of $\mathcal{P}_k'(q)$ in (10), by

$$\mathcal{P}_k'(q, \mathcal{M}) = \{p \in \mathcal{P}_k : \mathsf{cond}(q, \mathcal{M})\} \quad \text{with} \quad \mathsf{cond}(q, \mathcal{M}) = \begin{cases} p = q & \text{if } \mathcal{M} \text{ is FI or MAB}, \\ p \geq q/(2k) & \text{if } \mathcal{M} \text{ is PM-local}. \end{cases}$$

We then consider the following optimization problem, which can be seen as a slight generalization of the approach developed in [46]:

$$(p_t, G_t) = \underset{p \in \mathcal{P}_k'(q_t, \mathcal{M}), \, G \in \mathcal{H}}{\arg\min} \mathsf{ST}(p, G; q_t, \eta_t), \tag{12}$$

where the feasible region $\mathcal{P}_k$ of $p$ is replaced with $\mathcal{P}_k'(q, \mathcal{M})$. We define the optimal value of (12) by $\mathrm{opt}_q'(\eta, \mathcal{M})$ and its truncation at round $t$ by $V_t'(\mathcal{M}) = \max\{0, \mathrm{opt}_{q_t}'(\eta_t, \mathcal{M})\}$. We will abbreviate $\mathcal{M}$ when it is clear from a context. The following proposition shows that the component of the regret in (9) can be made small enough even if the feasible region is restricted to $\mathcal{P}_k'(q, \mathcal{M}) \subset \mathcal{P}_k$.

**Proposition 1.** *Let $\mathcal{M}$ be an underlying model. If $\mathcal{M}$ is FI, MAB, or PM-local with $\eta \leq 1/(2mk^2)$,*

$$\mathrm{opt}_*'(\eta) := \sup_{q \in \mathcal{P}_k} \mathrm{opt}_q'(\eta) \leq \bar{V}(\mathcal{M}) := \begin{cases} 1/2 & \text{if } \mathcal{M} \text{ is FI} \\ k/2 & \text{if } \mathcal{M} \text{ is MAB} \\ 3m^2 k^3 & \text{if } \mathcal{M} \text{ is PM-local}. \end{cases}$$

One can immediately obtain this result by following the same lines as [32, Propositions 11 and 12] and [46, Lemma 5].

## B.2 Algorithm

We use the negative Shanon entropy regularizer $\Phi_t = \frac{1}{\eta_t} \psi^{\mathsf{nS}}$ for (1) with learning rate $\eta_t = 1/\beta_t$ with

$$\beta_1 = B\sqrt{\frac{\log(1 + T)}{\log k}} \quad \text{and} \quad \beta_{t+1} = \beta_t + \frac{c_1 V_t'}{\sqrt{\bar{V} h_1 + \sum_{s=1}^{t-1} V_s' h_{s+1}}}, \tag{13}$$

where $B = 1/2$ for FI, $B = k/2$ for MAB, and $B = 2mk^2$ for PM-local, which corresponds to the learning rate in Definition 2 with $h_t \leftarrow H(q_t)$, $z_t \leftarrow V_t'$, $\bar{z}_t \leftarrow 0$, and $c_2 \leftarrow 0$. Algorithm 1 summarizes the proposed algorithm.

## B.3 Main result

Let $r_{\mathcal{M}}$ be $1$ if $\mathcal{M}$ is FI or MAB, and $2k$ if $\mathcal{M}$ is PM-local. Then we have the following bound.

**Corollary 6** (Extended version of Corollary 5)**.** *Let $\mathcal{M}$ be FI, MAB, or PM-local. Then the above algorithm with $c_1 = \sqrt{\log(1+T)/2}$ (Algorithm 1) achieves*

$$\mathsf{Reg}_T \leq \mathbb{E}\left[\sqrt{2\sum_{t=1}^{T} V_t' \log(k) \log(1+T)}\right] + O(B\sqrt{\log(k)\log(T)})$$

*in the adversarial regime, and*

$$\mathsf{Reg}_T = O\left(\frac{r_{\mathcal{M}} \bar{V} \log(T) \log(kT)}{\Delta_{\min}} + \sqrt{\frac{Cr_{\mathcal{M}} \bar{V} \log(T) \log(kT)}{\Delta_{\min}}} + B\sqrt{\log(k)\log(T)}\right)$$

*in the adversarial regime with a $(\Delta, C, T)$ self-bounding constraint.*

The bound for the adversarial regime with a self-bounding constraint with $C = 0$ yields the bound in the stochastic regime, which is optimal up to logarithmic factors in FI and MAB, and has the same order as the bounds in [46, Theorem 6].

The bound for the adversarial regime has a form similar to [32] and is game-dependent in the sense that it can be bounded by the empirical difficulty $V_t'$ of the current game. In addition, we can also obtain the worst-case bound by replacing $V_t'$ with its upper bound $\bar{V}$. This bound is optimal up to $\log(T)$ factor in FI and $\log(k)\log(T)$ factor in MAB, and is a factor of $\sqrt{\log T}$ worse than the best known bound in [32], which can be seen as the cost for the BOBW guarantee (see also Table 2).

## C  Additional related work

This appendix provides additional related work that could not be included in the main body due to the numerous studies related to this paper.

**Multi-armed bandits**    In the stochastic regime, it is known that the optimal regret is approximately expressed as $\mathsf{Reg}_T = O(k \log T/\Delta_{\min})$ [29]. In the adversarial regime (a.k.a. non-stochastic regime), it is known that the Online Mirror Descent (OMD) framework with the (negative) Tsallis entropy regularizer achieves $O(\sqrt{kT})$ regret bounds [1, 4], which match the lower bound of $\Omega(\sqrt{kT})$ [5].

**Data-dependent bound**    In the adversarial MAB, algorithms with various data-dependent regret bounds have been developed. Typical examples of such bounds are first-order bounds dependent on the cumulative loss and second-order bounds depending on sample variances in losses. Allenberg et al. [2] provided an algorithm with a first-order regret bound of $O(\sqrt{kL^* \log k})$ for $L^* = \min_{a \in \mathcal{A}} \sum_{t=1}^{T} \langle \ell_t, a \rangle$. Second-order regret bounds have been shown in some studies, *e.g.,* [10, 19, 49], In particular, Bubeck et al. [10] provided the regret bound of $O(\sqrt{Q_2 \log k})$ for $Q_2 = \sum_{t=1}^{T} \|\ell_t - \bar{\ell}\|^2$. Other examples of data-dependent bounds include path-length bounds in the form of $O(\sqrt{kV_1 \log T})$ for $V_1 = \sum_{t=1}^{T-1} \|\ell_t - \ell_{t+1}\|_1$ as well as a sparsity-dependent bound, which have been investigated [10, 11, 27, 49, 51].

**Sparsity-dependent bound**    The study on a sparsity-dependent bound was initiated by Kwon and Perchet [27], who showed that when $\ell_t \in [0,1]^k$, the OMD with Tsallis entropy can achieve the bound of $\mathsf{Reg}_T \leq 2\sqrt{e}\sqrt{sT \log(k/s)}$ and prove the matching (up to logarithmic factor) lower bound of $\mathsf{Reg}_T = \Omega(\sqrt{sT})$ when $T \geq k^3/(4s^2)$. Bubeck et al. [10] also showed that OMD with a hybrid-regularizer consisting of the Shannon entropy and a log-barrier can achieve $\mathsf{Reg}_T \leq 10\sqrt{L_2 \log k} + 20k \log T$ when $\ell_t \in [-1,1]^k$. Zheng et al. [51] investigated the sparse MAB problem in the context of the switching regret. Although their result is not directly related to our study, they show that sparsity is useful in some cases. Note that all of these algorithms assume the knowledge of the sparsity level and do not have a BOBW guarantee. The study to exploit the sparsity was investigated also in the stochastic regime by Kwon et al. [28]. However, they define the sparsity level $s$ by the number of arms with rewards larger than or equal to 0 (*i.e.,* losses smaller than or equal to 0), and hence the definition of sparsity is different from that in our paper.

**Adaptive learning rate**    The stability-dependent learning rate is quite ubiquitous (see [38] and the references therein). To our knowledge, the literature on the penalty-dependent bound is quite scarce in bandits and considered in the context of BOBW algorithms [22, 46], both of which consider the Shannon entropy regularizer.

**Best-of-both-worlds**    Since the seminal study by Bubeck and Slivkins [9], BOBW algorithms have been developed for many online decision-making problems. Although they have been investigated mainly in the context of an MAB [43, 53], other settings have also been investigated, [18, 24, 49], to name a few.

FTRL and OMD are now one of the most common approaches to achieving a BOBW guarantee owing to the usefulness of the self-bounding technique [18, 49, 53], while the first [9] and earlier work [43, 44] on BOBW do not rely on the technique. Most of the recent algorithms beyond the MAB are also based on FTRL (to name a few, [24, 42, 49]).

Our BOBW algorithm with the sparsity-dependent bound can be seen as one of the studies that aim to achieve BOBW and data-dependent bound simultaneously. There is not so much existing research, and we are only aware of [21, 23, 47, 49]. They consider first-, second-order, and path-length bound, and we are the first to investigate the sparsity-dependent bound in this line of work.

**Log-barrier regularizer and hybrid regularizer**    The log-barrier regularizer has been used in various studies (to name a few, [14, 16, 35, 40, 49]). The time-invariant log-barrier (a.k.a. constant amount of log-barrier [33]), whose properties are extensively exploited in this paper, was invented by Bubeck et al. [10] and has been used in several subsequent studies [33, 51].

**Partial monitoring**    Starting from the work by Rustichini [41], PM has been investigated in many works in the literature [7, 12, 39]. It is known that all PM games can be classified into four classes based on their minimax regrets [7, 30]. In particular, all PM games fall into trivial, easy, hard, and hopeless games, for which its minimax regrets is $0$, $\Theta(\sqrt{T})$, $\Theta(T^{2/3})$, and $\Theta(T)$, respectively. PM has also been investigated in both the adversarial and stochastic regimes as for MAB. In the stochastic regime, there are relatively small amount of works [8, 26, 45, 48], some of which are proven to achieve an instance-dependent $O(\log T)$ regrets for locally or globally observable games. In the adversarial regime, since the development of the FeedExp3 algorithm [12, 39], many algorithms achieving the minimax optimal regret have been developed [6, 7, 15, 31].

## D    Proof of Theorem 1

*Proof.*   We first prove (5) in Part (I).

**Penalty term**    First, we consider the penalty term $\sum_{t=1}^{T} \left( \frac{1}{\eta_{t+1}} - \frac{1}{\eta_t} \right) h_{t+1}$. By the definition of $\beta_t$ in (4),

$$
\sum_{t=1}^{T} \left( \frac{1}{\eta_{t+1}} - \frac{1}{\eta_t} \right) h_{t+1} = \sum_{t=1}^{T} (\beta_{t+1} - \beta_t) h_{t+1} = \sum_{t=1}^{T} \frac{c_1 z_t h_{t+1}}{\sqrt{c_2 + \bar{z}_t h_1 + \sum_{s=1}^{t-1} z_s h_{s+1}}}
$$

$$
\leq c_1 \sum_{t=1}^{T} \frac{z_t h_{t+1}}{\sqrt{\sum_{s=1}^{t} z_s h_{s+1}}} \leq c_1 \int_0^{\sum_{t=1}^{T} z_t h_{t+1}} \frac{1}{\sqrt{x}} \, \mathrm{d}x = 2c_1 \sqrt{\sum_{t=1}^{T} z_t h_{t+1}}, \tag{14}
$$

where the first inequality follows from $\bar{z}_t h_1 \geq z_t h_{t+1}$ and the second inequality follows by Lemma 8 given in Appendix J.

**Stability term**    Next, we consider the stability term $\sum_{t=1}^{T} \eta_t z_t$. Using the definition of $\beta_t$ in (4) and defining $U_t = \sqrt{c_2 + \bar{z}_t h_1 + \sum_{s=1}^{t-1} z_s h_{s+1}}$ for $t \in \{0\} \cup [T]$, we bound $\beta_t$ from below as

$$
\beta_t = \beta_1 + \sum_{u=1}^{t-1} \frac{c_1 z_u}{\sqrt{c_2 + \bar{z}_u h_1 + \sum_{s=1}^{u-1} z_s h_{s+1}}} = \beta_1 + \sum_{u=1}^{t-1} \frac{c_1 z_u}{U_u} \geq \beta_1 + \frac{c_1}{U_T} \sum_{u=1}^{t-1} z_u, \tag{15}
$$

where the inequality follows since $(U_t)$ is non-decreasing. Using the last inequality, we can bound $\sum_{t=1}^{T} \eta_t z_t$ as

$$\sum_{t=1}^{T} \eta_t z_t = 2 \sum_{t=1}^{T} \frac{z_t}{\beta_t + \beta_t} \leq 2 \sum_{t=1}^{T} \frac{z_t}{\beta_t + \beta_1 + \frac{c_1}{U_T} \sum_{s=1}^{t-1} z_s} = \frac{2U_T}{c_1} \sum_{t=1}^{T} \frac{z_t}{\frac{U_T}{c_1} (\beta_t + \beta_1) + \sum_{s=1}^{t-1} z_s} . \tag{16}$$

Since we have $\frac{U_T}{c_1} (\beta_1 + \beta_t) \geq \frac{\sqrt{c_2 + \bar{z}_t h_1}}{c_1} (\beta_1 + \beta_t) \geq \varepsilon + z_t$ by the assumption, a part of the last inequality is further bounded as

$$\sum_{t=1}^{T} \frac{z_t}{\frac{U_T}{c_1} (\beta_t + \beta_1) + \sum_{s=1}^{t-1} z_s} \leq \sum_{t=1}^{T} \frac{z_t}{\varepsilon + \sum_{s=1}^{t} z_s} \leq \int_{\varepsilon}^{\varepsilon + \sum_{t=1}^{T} z_t} \frac{1}{x} \, \mathrm{d}x \leq \log \left( 1 + \sum_{t=1}^{T} \frac{z_t}{\varepsilon} \right), \tag{17}$$

where the second inequality follows by Lemma 8. Combining (16) and (17) yields

$$\sum_{t=1}^{T} \eta_t z_t \leq \frac{2U_T}{c_1} \log \left( 1 + \sum_{t=1}^{T} \frac{z_t}{\varepsilon} \right) = \frac{2}{c_1} \log \left( 1 + \sum_{t=1}^{T} \frac{z_t}{\varepsilon} \right) \sqrt{c_2 + \bar{z}_T h_1 + \sum_{t=1}^{T} z_t h_{t+1}}. \tag{18}$$

Combining (14) and (18) completes the proof of (5) in Part (I).

We next prove Part (II). For the penalty term, setting $h_t = h_1$ for all $t \in [T]$ in (14) gives

$$\sum_{t=1}^{T} \left( \frac{1}{\eta_{t+1}} - \frac{1}{\eta_t} \right) h_{t+1} \leq 2c_1 \sqrt{h_1 \sum_{t=1}^{T} z_t} .$$

For the stability term, since there exists $a > 0$ such that $\beta_t \geq \frac{ac_1}{\sqrt{h_1}} \sqrt{\sum_{s=1}^{t} z_s}$ for any $t \in [T]$ by the assumption,

$$\sum_{t=1}^{T} \eta_t z_t = \sum_{t=1}^{T} \frac{z_t}{\beta_t} \leq \frac{\sqrt{h_1}}{ac_1} \sum_{t=1}^{T} \frac{z_t}{\sqrt{\sum_{s=1}^{t} z_s}} \leq \frac{2}{ac_1} \sqrt{h_1 \sum_{t=1}^{T} z_t} .$$

Summing up the above arguments completes the proof of Part (II). $\qquad \square$

## E    Basic facts to bound stability terms

Here, we introduce basic facts, which are useful to bound the stability term. We have

$$\xi(x) := \exp(-x) + x - 1 \leq \begin{cases} \frac{1}{2} x^2 & \text{for } x \geq 0 \\ x^2 & \text{for } x \geq -1 , \end{cases} \tag{19}$$

$$\zeta(x) := x - \log(1 + x) \leq x^2 \quad \text{for } x \in \left[ -\frac{1}{2}, \frac{1}{2} \right] . \tag{20}$$

We also have the following inequalities for $\phi^{\mathsf{nS}}(x) = x \log x$ and $\phi^{\mathsf{LB}}(x) = \log(1/x)$, which are components of the negative Shannon entropy and log-barrier function:

$$\max_{y \in \mathbb{R}} \left\{ a(x - y) - D_{\phi^{\mathsf{nS}}}(y, x) \right\} = x \xi(a) \qquad \text{for } a \in \mathbb{R} , \tag{21}$$

$$\max_{y \in \mathbb{R}} \left\{ a(x - y) - D_{\phi^{\mathsf{LB}}}(y, x) \right\} = \zeta(ax) \qquad \text{for } a \geq -\frac{1}{x} . \tag{22}$$

It is easy to prove these facts by the standard calculus and you can find the proofs of (21) and (22) in Lemma 15 of Tsuchiya et al. [46] and Lemma 5 of Ito et al. [23], respectively.

# F   Proof of Corollary 2

Let $\text{Reg}_T(a) = \mathbb{E}\left[\sum_{t=1}^{T}(\ell_{tA_t} - \ell_{ta})\right]$ for $a \in [k]$. Here we provide the complete proof of Corollary 2.

*Proof of Corollary 2.* Fix $i^* \in [k]$. Since $p_t = (1 - \gamma)q_t + \frac{\gamma}{k}\mathbf{1}$, it holds that

$$
\begin{aligned}
\text{Reg}_T(i^*) &= \mathbb{E}\left[\sum_{t=1}^{T}\ell_{tA_t} - \sum_{t=1}^{T}\ell_{ti^*}\right] = \mathbb{E}\left[\sum_{t=1}^{T}\langle \ell_t, p_t - e_{i^*}\rangle\right] \\
&= \mathbb{E}\left[\sum_{t=1}^{T}\langle \ell_t, q_t - e_{i^*}\rangle\right] + \mathbb{E}\left[\gamma\sum_{t=1}^{T}\left\langle \ell_t, \frac{1}{k}\mathbf{1} - q_t\right\rangle\right] \leq \mathbb{E}\left[\sum_{t=1}^{T}\langle \widehat{y}_t, q_t - e_{i^*}\rangle\right] + \gamma T\,,
\end{aligned}
$$

where the last inequality follows by $\mathbb{E}[\widehat{y}_t \mid q_t] = \ell_t$ and the Cauchy-Schwarz inequality. Then, using the standard analysis of the FTRL described in Section 3, the first term in the last inequality is bounded as

$$
\sum_{t=1}^{T}\langle \widehat{y}_t, q_t - e_{i^*}\rangle \leq \sum_{t=1}^{T}\left(\frac{1}{\eta_{t+1}} - \frac{1}{\eta_t}\right)H(q_{t+1}) + \frac{H(q_1)}{\eta_1} + \sum_{t=1}^{T}\left(\langle q_t - q_{t+1}, \widehat{y}_t\rangle - D_{\Phi_t}(q_{t+1}, q_t)\right)\,.
$$

By (19) and (21) given in Appendix E, the stability term $\langle q_t - q_{t+1}, \widehat{y}_t\rangle - D_{\Phi_t}(q_{t+1}, q_t)$ in the last inequality is bounded as

$$
\begin{aligned}
\langle q_t - q_{t+1}, \widehat{y}_t\rangle - D_{\Phi_t}(q_{t+1}, q_t) &= \langle q_t - q_{t+1}, \widehat{y}_t\rangle - \frac{1}{\eta_t}D_{\psi^{\text{ns}}}(q_{t+1}, q_t) \\
&= \sum_{i=1}^{k}\left(\widehat{y}_{ti}(q_{ti} - q_{t+1,i}) - \frac{1}{\eta_t}D_{\phi^{\text{ns}}}(q_{t+1,i}, q_{ti})\right) \\
&\leq \sum_{i=1}^{k}\frac{1}{\eta_t}q_{ti}\,\xi\left(\eta_t\widehat{y}_{ti}\right) \leq \frac{1}{2}\eta_t\sum_{i=1}^{k}q_{ti}\widehat{y}_{ti}^2 \leq \eta_t\omega_t\,,
\end{aligned}
$$

where the first inequality follows from (21), the second inequality follows by (19) with $\widehat{y}_t \geq 0$, and the last inequality holds since $\sum_{i=1}^{k}q_{ti}\widehat{y}_{ti}^2 \leq \sum_{i=1}^{k}2p_{ti}\widehat{y}_{ti}^2 = 2\omega_t$.

We will confirm that the assumptions for Part (II) of Theorem 1 are indeed satisfied. Using the definition of $\beta_t$ in (6), We have

$$
\begin{aligned}
\beta_t &= \beta_1 + \frac{1}{\sqrt{h_1}}\sum_{u=1}^{t-1}\frac{c_1\omega_u}{\sqrt{\frac{k}{\gamma} + \sum_{s=1}^{u-1}\omega_s}} = \beta_1 + \frac{2c_1}{\sqrt{h_1}}\sum_{u=1}^{t-1}\frac{\omega_u}{\sqrt{\frac{k}{\gamma} + \sum_{s=1}^{u-1}\omega_s} + \sqrt{\frac{k}{\gamma} + \sum_{s=1}^{u-1}\omega_s}} \\
&\geq \beta_1 + \frac{2c_1}{\sqrt{h_1}}\sum_{u=1}^{t-1}\frac{\omega_u}{\sqrt{\frac{k}{\gamma} + \sum_{s=1}^{u}\omega_s} + \sqrt{\frac{k}{\gamma} + \sum_{s=1}^{u-1}\omega_s}} \\
&= \beta_1 + \frac{2c_1}{\sqrt{h_1}}\sum_{u=1}^{t-1}\left(\sqrt{\frac{k}{\gamma} + \sum_{s=1}^{u}\omega_s} - \sqrt{\frac{k}{\gamma} + \sum_{s=1}^{u-1}\omega_s}\right) \\
&= \beta_1 + \frac{2c_1}{\sqrt{h_1}}\left(\sqrt{\frac{k}{\gamma} + \sum_{s=1}^{t-1}\omega_s} - \sqrt{\frac{k}{\gamma}}\right) \geq \frac{2c_1}{\sqrt{h_1}}\sqrt{\sum_{s=1}^{t}\omega_s}\,,
\end{aligned}
$$

where the last inequality follows since $\beta_1 = \frac{2c_1}{\sqrt{h_1}}\sqrt{\frac{k}{\gamma}}$ and $\frac{k}{\gamma} \geq \omega_t$. Hence, stability condition (S2) in Theorem 1 is satisfied with $a = 2$, and one can see that the other assumptions are trivially satisfied. Hence, by Part (II) of Theorem 1,

$$
\sum_{t=1}^{T}\langle \widehat{y}_t, q_t - e_{i^*}\rangle \leq \sum_{t=1}^{T}\left(\frac{1}{\eta_{t+1}} - \frac{1}{\eta_t}\right)H(q_{t+1}) + \sum_{t=1}^{T}\eta_t\omega_t + \frac{H(q_1)}{\eta_1} \leq 2\left(c_1 + \frac{1}{2c_1}\right)\sqrt{h_1\sum_{t=1}^{T}\omega_t} + \frac{\log k}{\eta_1}\,,
$$

where in the last inequality we used $h_1 \le \log k$. Now,

$$\mathbb{E}\left[\sqrt{\sum_{t=1}^{T}\omega_t}\right] \le \sqrt{\sum_{t=1}^{T}\mathbb{E}[\omega_t]} = \sqrt{\sum_{t=1}^{T}\mathbb{E}\left[\frac{\ell_{tA_t}^2}{p_{tA_t}}\right]} = \sqrt{\sum_{t=1}^{T}\sum_{i=1}^{k}\ell_{ti}^2} = \sqrt{\sum_{t=1}^{T}\|\ell_t\|_2^2} = \sqrt{L_2}\,.$$

Summing up the above arguments and setting $c_1 = 1/\sqrt{2}$, we have

$$\mathsf{Reg}_T \le 2\sqrt{2}\sqrt{L_2 \log k} + \frac{\log k}{\eta_1} + \gamma T = 2\sqrt{2}\sqrt{L_2 \log k} + (\sqrt{2}+1)(kT\log k)^{1/3}\,,$$

which completes the proof of Corollary 2. $\qquad\qquad\square$

## G  Proof of Corollary 3

We first prove Lemma 3.

*Proof of Lemma 3.* Recall that $\Phi_t(p) = \frac{1}{\eta_t}\psi^{\mathsf{nS}}(p) + 2\delta\psi^{\mathsf{LB}}(p)$. Since $D_{\Phi_t} = \frac{1}{\eta_t}D_{\psi^{\mathsf{nS}}} + 2\delta D_{\psi^{\mathsf{LB}}}$ and $D_{\psi^{\mathsf{nS}}}(x,y) = \sum_{i=1}^{k}D_{\phi^{\mathsf{nS}}}(x_i,y_i)$ and $D_{\psi^{\mathsf{LB}}}(x,y) = \sum_{i=1}^{k}D_{\phi^{\mathsf{LB}}}(x_i,y_i)$, we can bound the stability term as

$$\langle q_t - q_{t+1}, \widehat{y}_t\rangle - D_{\Phi_t}(q_{t+1}, q_t) \le \langle q_t - q_{t+1}, \widehat{y}_t\rangle - \max\left\{\frac{1}{\eta_t}D_{\psi^{\mathsf{nS}}}(q_{t+1}, q_t), 2\delta D_{\psi^{\mathsf{LB}}}(q_{t+1}, q_t)\right\}$$

$$\le \sum_{i=1}^{k}\left(\widehat{y}_{ti}(q_{ti} - q_{t+1,i}) - \max\left\{\frac{1}{\eta_t}D_{\phi^{\mathsf{nS}}}(q_{t+1,i}, q_{ti}), 2\delta D_{\phi^{\mathsf{LB}}}(q_{t+1,i}, q_{ti})\right\}\right)$$

$$\le \sum_{i=1}^{k}\min\left\{\frac{1}{\eta_t}q_{ti}\,\xi\,(\eta_t\widehat{y}_{ti})\,, 2\delta\,\zeta\left(\frac{1}{2\delta}q_{ti}\widehat{y}_{ti}\right)\right\}\,, \tag{23}$$

where in the last inequality we used (21) and (22) with

$$\frac{\widehat{y}_{ti}}{2\delta} \ge -\frac{1}{2\delta p_{ti}} \ge -\frac{1}{2\delta(q_{ti}/\delta)} \ge -\frac{1}{q_{ti}}\,,$$

where the first inequality follows by the definition of $\widehat{y}_t$ and the second inequality follows by $p_{ti} \ge q_{ti}/\delta$.

Next, we will prove that for any $i \in [k]$,

$$\min\left\{\frac{1}{\eta_t}q_{ti}\,\xi\,(\eta_t\widehat{y}_{ti})\,, 2\delta\,\zeta\left(\frac{1}{2\delta}q_{ti}\widehat{y}_{ti}\right)\right\} \le \delta\eta_t\frac{\ell_{ti}^2}{p_{ti}}\min\left\{1, \frac{p_{ti}}{2\eta_t}\right\}\mathbb{1}[A_t = i]\,. \tag{24}$$

Fix $i \in [k]$. By $q_{ti} \le \delta p_{ti}$,

$$\frac{1}{2\delta}q_{ti}\widehat{y}_{ti} = \frac{1}{2}p_{ti}\widehat{y}_{ti} \le \frac{1}{2}\,.$$

Using this and $\zeta(x) \le x^2$ for $x \in [-\frac{1}{2}, \frac{1}{2}]$ in (20), we have for any $p_{ti} \in [0, 1]$ that

$$2\delta\,\zeta\left(\frac{1}{2\delta}q_{ti}\widehat{y}_{ti}\right) \le 2\delta\left(\frac{1}{2\delta}q_{ti}\widehat{y}_{ti}\right)^2 \le \frac{\delta}{2}\ell_{ti}^2\mathbb{1}[A_t = i]\,, \tag{25}$$

where in the last inequality we used $q_{ti} \le \delta p_{ti}$. In particular, when $p_{ti} \le \eta_t$, *i.e.*, the probability of selecting arm $i$ is small to some extent, the last inequality can be further bounded as

$$2\delta\,\zeta\left(\frac{1}{2\delta}q_{ti}\widehat{y}_{ti}\right) \le \frac{\eta_t}{p_{ti}}\frac{\delta}{2}\ell_{ti}^2\mathbb{1}[A_t = i] \le \delta\eta_t\frac{\ell_{ti}^2}{p_{ti}}\mathbb{1}[A_t = i]\,. \tag{26}$$

On the other hand when $p_{ti} > \eta_t$, we have $\eta_t\widehat{y}_{ti} \ge -1$. Hence, by the inequality $\xi(x) \le x^2$ for $x \ge -1$ in (19),

$$\frac{1}{\eta_t}q_{ti}\xi\,(\eta_t\widehat{y}_{ti}) \le \frac{1}{\eta_t}\delta p_{ti}(\eta_t\widehat{y}_{ti})^2 = \delta\eta_t\frac{\ell_{ti}^2}{p_{ti}}\mathbb{1}[A_t = i]\,. \tag{27}$$

Hence, combining (25), (26), and (27) completes the proof of (24). Finally, by combining (23) and (24) we completes the proof of Lemma 3. $\qquad\square$

**Remark.** When $\ell_t$ can be negative, the Shannon entropy regularizer alone cannot bound the stability term if the arm selection probability is small, *i.e.*, $p_{ti} \leq \eta_t$. Introducing a time-invariant log-barrier regularizer enables us to bound the stability term even when the arm selection probability is small. This idea was proposed by Bubeck et al. [10], who analyzed the variation of arm selection probability for the change of cumulative losses. Unlike their analysis, our proof directly analyses the stability term, enabling us to obtain the tighter regret bound. More importantly, we will utilize the property $\nu_t \leq O(1/\eta_t)$ many times, which directly follows from Lemma 3, in the subsequent sections to prove the BOBW guarantee with the sparsity-dependent bound.

Now, we are ready to prove Corollary 3.

*Proof of Corollary 3.* Fix $i^* \in [k]$. Define $p^* \in \mathcal{P}_k$ by

$$p^* = \left(1 - \frac{k}{T}\right) e_{i^*} + \frac{1}{T}\mathbf{1}.$$

Then, using the definition of the algorithm,

$$
\begin{aligned}
\mathrm{Reg}_T(i^*) = \mathbb{E}\left[\sum_{t=1}^T \ell_{tA_t} - \sum_{t=1}^T \ell_{ti^*}\right] &= \mathbb{E}\left[\sum_{t=1}^T \langle \ell_t, p_t - e_{i^*}\rangle\right] \\
&= \mathbb{E}\left[\sum_{t=1}^T \langle \ell_t, p_t - p^*\rangle\right] + \mathbb{E}\left[\sum_{t=1}^T \langle \ell_t, p^* - e_{i^*}\rangle\right] \\
&\leq \mathbb{E}\left[\sum_{t=1}^T \langle \widehat{y}_t, p_t - p^*\rangle\right] + k,
\end{aligned}
$$

where the inequality follows from the definition of $p^*$ and the Cauchy-Schwarz inequality. By the standard analysis of the FTRL, described in Section 3,

$$
\sum_{t=1}^T \langle \widehat{y}_t, p_t - p^*\rangle \leq \sum_{t=1}^T \Big(\Phi_t(p_{t+1}) - \Phi_{t+1}(p_{t+1})\Big) + \Phi_{t+1}(p^*) - \Phi_1(p_1) + \sum_{t=1}^T \Big(\langle p_t - p_{t+1}, \widehat{y}_t\rangle - D_{\Phi_t}(p_{t+1}, p_t)\Big).
$$

For the penalty term, since $\Phi_t(p) = \frac{1}{\eta_t}\psi^{\mathsf{nS}}(p) + 2\psi^{\mathsf{LB}}(p)$,

$$
\begin{aligned}
\sum_{t=1}^T &\Big(\Phi_t(p_{t+1}) - \Phi_{t+1}(p_{t+1})\Big) + \Phi_{t+1}(p^*) - \Phi_1(p_1) \\
&\leq \sum_{t=1}^T \left(\frac{1}{\eta_{t+1}} - \frac{1}{\eta_t}\right) H(p_{t+1}) + \frac{H(p_1)}{\eta_1} + 2\sum_{i=1}^k \log\left(\frac{1}{p_i^*}\right) \\
&\leq \sum_{t=1}^T \left(\frac{1}{\eta_{t+1}} - \frac{1}{\eta_t}\right) H(p_{t+1}) + \frac{\log k}{\eta_1} + 2k\log T,
\end{aligned}
$$

where in the last inequality we used the fact that $p_i^* \geq 1/T$ for all $i \in [k]$.

For the stability term, by Lemma 3 with $\delta = 1$ (since $p_t = q_t$),

$$
\sum_{t=1}^T \Big(\langle p_t - p_{t+1}, \widehat{y}_t\rangle - D_{\Phi_t}(p_{t+1}, p_t)\Big) \leq \sum_{t=1}^T \eta_t \nu_t.
$$

We will confirm that the assumptions for Part (II) of Theorem 1 are indeed satisfied. By the definition of the learning rate in (7),

$$
\begin{aligned}
\beta_t = \beta_1 + \sum_{u=1}^{t-1} \frac{c_1 \nu_u}{\sqrt{h_1}\sqrt{\sum_{s=1}^u \nu_s}} &\geq \beta_1 + \frac{c_1}{\sqrt{h_1}}\sum_{u=1}^{t-1} \frac{\nu_u}{\sqrt{\sum_{s=1}^u \nu_s} + \sqrt{\sum_{s=1}^{u-1}\nu_s}} \\
&\geq \beta_1 + \frac{c_1}{\sqrt{h_1}}\sum_{u=1}^{t-1}\left(\sqrt{\sum_{s=1}^u \nu_s} - \sqrt{\sum_{s=1}^{u-1}\nu_s}\right) = \beta_1 + \frac{c_1}{\sqrt{h_1}}\sqrt{\sum_{s=1}^{t-1}\nu_s}.
\end{aligned}
$$

Using this inequality, $\beta_t$ is bounded from below as

$$2\beta_t = \beta_t + \beta_t \geq 2\nu_t + \beta_1 + \frac{c_1}{\sqrt{h_1}}\sqrt{\sum_{s=1}^{t-1}\nu_s} \geq 2\sqrt{2\beta_1\nu_t} + \frac{c_1}{\sqrt{h_1}}\sqrt{\sum_{s=1}^{t-1}\nu_s} \geq \frac{c_1}{\sqrt{h_1}}\sqrt{\sum_{s=1}^{t}\nu_s},$$

where the first inequality follows by $\nu_t \leq \beta_t/2$ and the above inequality, the second inequality follows by the AM-GM inequality, and the last inequality follows from $2\sqrt{2\beta_1} \geq \frac{c_1}{\sqrt{h_1}}$ and $\sqrt{x} + \sqrt{y} \geq \sqrt{x+y}$ for $x, y \geq 0$. Dividing the both sides by 2, we can see stability condition (S2) in Theorem 1 is satisfied with $a = 1/2$. One can also see that the other assumptions are trivially satisfied. Hence, by Part (II) of Theorem 1,

$$\sum_{t=1}^{T}\left(\frac{1}{\eta_{t+1}} - \frac{1}{\eta_t}\right)H(p_{t+1}) + \sum_{t=1}^{T}\eta_t\nu_t \leq 2\left(c_1 + \frac{2}{c_1}\right)\sqrt{h_1\sum_{t=1}^{T}\nu_t}.$$

Using the last inequality with $\mathbb{E}\left[\sqrt{\sum_{t=1}^{T}\nu_t}\right] \leq \mathbb{E}\left[\sqrt{\sum_{t=1}^{T}\omega_t}\right] \leq \sqrt{L_2}$, and setting $c_1 = \sqrt{2}$, we have

$$\mathsf{Reg}_T(i^*) \leq \mathbb{E}\left[2\left(c_1 + \frac{2}{c_1}\right)\sqrt{h_1\sum_{t=1}^{T}\nu_t} + \frac{\log k}{\eta_1} + 2k\log T + k\right]$$

$$\leq 4\sqrt{2}\sqrt{L_2\log k} + 2k\log T + k + \frac{1}{4},$$

which completes the proof of Corollary 3. $\qquad\square$

# H   Proof of results in Section 5.2

Appendix H.1 provides preliminary results, which will be used to quantify the difference between $q_t$ and $q_{t+1}$ in Appendix H.2 and will be used to prove the continuity of $F_t$ in Appendix H.3. Appendix H.2 proves Theorem 4 and Appendix H.3 discusses the bisection method to compute $\beta_{t+1}$.

## H.1   Some stability results

Before proving Theorem 4, we prove several important lemmas. Consider the following three optimization problems:

$$p \in \arg\min_{p' \in \mathcal{P}_k} \langle L - \xi e_1, p'\rangle + \beta\psi(p'),$$
$$q \in \arg\min_{q' \in \mathcal{P}_k} \langle L, q'\rangle + \beta\psi(q'), \tag{28}$$
$$r \in \arg\min_{r' \in \mathcal{P}_k} \langle L, r'\rangle + \beta'\psi'(r')$$

with $L \in \mathbb{R}_+^k$, $0 \leq \xi \leq \min_{i \in [k]} L_i$, and $\beta, \beta' > 0$ satisfying $\beta' \geq \beta$,

$$\psi(q) = \sum_{i=1}^{k}(q_i\log q_i - q_i) - \frac{c}{\beta}\sum_{i=1}^{k}\log q_i \quad\text{and}\quad \psi'(q) = \sum_{i=1}^{k}(q_i\log q_i - q_i) - \frac{c}{\beta'}\sum_{i=1}^{k}\log q_i$$

for $c > 0$. Note that the outputs of FTRL with $\psi(q)$ and with $-H(q) - (c/\beta)\sum_{i=1}^{k}\log q_i$ are identical since adding a constant to $\psi$ does not change the output of the above optimization problems.

In the following lemma, we investigate the relation between $q$ and $r$ in (28).

**Lemma 4.** *Consider $q$ and $r$ in (28). Then,*

$$r_i \leq q_i^{\beta/\beta'}. \tag{29}$$

*Proof.* From the KKT conditions there exist $\mu, \mu' \in \mathbb{R}$ such that

$$L + \beta \nabla \psi(q) + \mu \mathbf{1} = 0 \quad \text{and} \quad L + \beta' \psi'(r) + \mu' \mathbf{1} = 0 \,,$$

which implies, by $(\nabla \psi(q))_i = \log q_i - \frac{c}{\beta q_i}$, that

$$L_i + \beta \log q_i - \frac{c}{q_i} + \mu = 0 \quad \text{and} \quad \eta' L_i + \beta \log r_i - \frac{c}{r_i} + \mu' = 0 \tag{30}$$

for all $i \in [k]$. This is equivalent to

$$q_i = \exp\left(-\frac{1}{\beta}\left(L_i - \frac{c}{q_i} + \mu\right)\right) \quad \text{and} \quad r_i = \exp\left(-\frac{1}{\beta'}\left(L_i - \frac{c}{r_i} + \mu'\right)\right) \,.$$

Removing $L_i$ from these equalities yields that

$$r_i = q_i^{\beta/\beta'} \exp\left(\frac{c}{\beta'}\left(\frac{1}{r_i} - \frac{1}{q_i}\right)\right) \exp\left(\frac{1}{\beta'}(\mu - \mu')\right) \,. \tag{31}$$

We will prove $\frac{d\mu}{d\beta} > 0$. Taking derivative with respect to $\beta$ of (30), we have

$$\log q_i + \left(\frac{1}{q_i} + \frac{c}{q_i^2}\right)\frac{dq_i}{d\beta} + \frac{d\mu}{d\beta} = 0 \,.$$

Multiplying $\left(\frac{1}{q_i} + \frac{c}{q_i^2}\right)^{-1}$ and summing over $i \in [k]$ in the last equality, we have

$$-\left(\frac{1}{q_i} + \frac{c}{q_i^2}\right)^{-1}\log(1/q_i) + \sum_{i=1}^{k}\frac{dq_i}{d\beta} + \left(\frac{1}{q_i} + \frac{c}{q_i^2}\right)^{-1}\frac{d\mu}{d\beta} = 0 \,,$$

which with the fact $\sum_{i=1}^{k}\frac{dq_i}{d\beta} = 0$ implies $\frac{d\mu}{d\beta} > 0$. Hence, since $\beta \leq \beta'$ we have $\mu \leq \mu'$.

When $r_i \leq q_i$, it is obvious that we get $r_i \leq q_i^{\beta/\beta'}$.

When $r_i > q_i$, using (31) with the inequalities $\beta \leq \beta'$ and $\mu \leq \mu'$,

$$r_i = q_i^{\beta/\beta'} \exp\left(\frac{c}{\beta'}\left(\frac{1}{r_i} - \frac{1}{q_i}\right)\right) \exp\left(\frac{1}{\beta'}(\mu - \mu')\right) \leq q_i^{\beta/\beta'} \,,$$

which is the desired bound. $\qquad \square$

**Lemma 5.** *Consider $p$, $q$, and $r$ in (28). Then, under $\eta := 1/\beta \leq \frac{1}{15k}$, we have*

$$r_i \leq 3p_i^{\beta/\beta'} \,. \tag{32}$$

*Proof.* By Lemma 8 of Bubeck et al. [10] we have $q_i \leq 3p_i$ for all $i \in [k]$. Using this with Lemma 4, we have

$$r_i \leq q_i^{\beta/\beta'} \leq 3q_i^{\beta/\beta'} \,.$$

$\qquad \square$

## H.2 Proof of Theorem 4

In this section, we will provide the proof of Theorem 4. We first see that the ratio $\beta_t/\beta_{t+1}$ is close to one to some extent.

**Lemma 6.** *The learning rate $\beta_t$ in (8) satisfies*

$$1 - \frac{\beta_t}{\beta_{t+1}} \in (0, 1/10] \,.$$

*Proof.* Recall that $\beta_t = \beta_1 + \sum_{u=1}^{t-1} b_u$ with $b_u = \frac{c_1 \nu_u}{U_u}$ and $U_t = \sqrt{c_2 + \bar{z}_t h_1 + \sum_{s=1}^{t-1} z_s h_{s+1}}$ for $t \in \{0\} \cup [T]$. It suffices to show

$$\frac{\beta_t}{\beta_{t+1}} = \frac{\beta_t}{\beta_t + b_t} \geq \frac{9}{10} \Leftrightarrow \beta_t \geq 9 b_t \,.$$

This indeed follows since using $\nu_t \leq \beta_t/2$ we have

$$b_t = \frac{c_1 \nu_t}{\sqrt{81 c_1^2 + \sum_{s=1}^{t-1} z_s h_s + z_t h_{t+1}}} \leq \frac{c_1 \nu_t}{\sqrt{81 c_1^2}} = \frac{\nu_t}{9} \leq \frac{\beta_t}{9} \,.$$

$\square$

Finally, we are ready to prove one of the key lemmas for proving the BOBW regret bound with the sparsity-dependent bound. Recall that we have $p_t = \left(1 - \frac{k}{T}\right) q_t + \frac{1}{T} \mathbf{1}$ and $h_t = \frac{1}{1 - \frac{k}{T}} H(p_t)$. Using the result in Appendix H.1, we will show that $h_{t+1}$ is bounded in terms of $h_t$.

**Lemma 7.** *Suppose that $\beta_t$ is defined as* (8). *Then,*

$$h_{t+1} \leq 3 h_t + \frac{20k}{9} \left( \frac{\beta_{t+1}}{\beta_t} - 1 \right) \log \left( \frac{T}{k} \right) h_{t+1} \,.$$

*Proof.* Let us recall that $q_t$ and $q_{t+1}$ are defined as

$$q_t \in \arg\min_{q \in \mathcal{P}_k} \left\langle \sum_{s=1}^{t-1} \widehat{y}_s, \, q \right\rangle + \Phi_t(q) \quad \text{and} \quad q_{t+1} \in \arg\min_{q \in \mathcal{P}_k} \left\langle \sum_{s=1}^{t} \widehat{y}_s, \, q \right\rangle + \Phi_{t+1}(q) \,,$$

which corresponds to optimization problems (28) with $p = q_t$, $L = \sum_{s=1}^{t} \widehat{y}_s$, $\xi = \widehat{y}_{t A_t}$, $\psi = \Phi_t/\beta_t$, $\eta = 1/\beta_t$, $r = q_{t+1}$, $\psi' = \Phi_{t+1}/\beta_{t+1}$, and $\eta' = 1/\beta_{t+1}$.

Since $H$ is concave, by $p_{ti} = (1 - \frac{k}{T}) q_{ti} + \frac{1}{T}$ and Jensen's inequality we have

$$\left(1 - \frac{k}{T}\right) h_t = H(p_t) \geq \left(1 - \frac{k}{T}\right) H(q_t) + \frac{k}{T} H\left(\frac{1}{k}\mathbf{1}\right) \geq \left(1 - \frac{k}{T}\right) H(q_t) \,,$$

which implies $h_t \geq H(q_t)$. By Lemma 5 we also have $q_{t+1,i} \leq 3 q_{ti}^{\beta_t/\beta_{t+1}}$, which implies that

$$p_{t+1,i} = \left(1 - \frac{k}{T}\right) q_{t+1,i} + \frac{1}{T} \leq \left(1 - \frac{k}{T}\right) 3 q_{ti}^{\beta_t/\beta_{t+1}} + \frac{1}{T} \leq 6 p_{ti}^{\beta_t/\beta_{t+1}} \,.$$

The last inequality follows since when $\left(1 - \frac{k}{T}\right) 3 q_{ti}^{\beta_t/\beta_{t+1}} \leq \frac{1}{T}$,

$$\left(1 - \frac{k}{T}\right) 3 q_{ti}^{\beta_t/\beta_{t+1}} + \frac{1}{T} \leq \frac{2}{T} \leq 2 \left(\frac{1}{T}\right)^{\beta_t/\beta_{t+1}} \leq 2 \left( \left(1 - \frac{k}{T}\right) q_{ti} + \frac{1}{T} \right)^{\beta_t/\beta_{t+1}} = 2 p_{ti}^{\beta/\beta_{t+1}} \,,$$

and otherwise

$$\left(1 - \frac{k}{T}\right) 3 q_{ti}^{\beta_t/\beta_{t+1}} + \frac{1}{T} \leq 6 \left(1 - \frac{k}{T}\right)^{\beta_t/\beta_{t+1}} q_{ti}^{\beta_t/\beta_{t+1}} \leq 6 p_{ti}^{\beta/\beta_{t+1}} \,.$$

Using these inequalities, we have

$$h_{t+1} - 3 h_t = \frac{1}{1 - \frac{k}{T}} \left( H(p_{t+1}) - 3 H(p_t) \right)$$

$$\leq \frac{1}{1 - \frac{k}{T}} \left( H(p_t) + \langle \nabla H(p_t), p_{t+1} - p_t \rangle - 3 H(p_t) \right)$$

$$= \frac{1}{1 - \frac{k}{T}} \sum_{i=1}^{k} (p_{t+1,i} - 3 p_{ti}) \log \left( \frac{1}{p_{ti}} \right)$$

$$\leq \sum_{i=1}^{k} (q_{t+1,i} - 3 q_{ti}) \log \left( \frac{1}{p_{ti}} \right) \,, \tag{33}$$

where the first inequality follows by the concavity of $H$, the second inequality follows since $p_{t+1,i} - 3p_{ti} \leq \left(1 - \frac{k}{T}\right)(q_{t+1,i} - q_{ti})$. Defining $\mathfrak{Q}_t = \{i \in [k] : q_{t+1,i} - 3q_{ti} \geq 0\}$, (33) is further bounded as

$$
\begin{aligned}
\sum_{i=1}^{k}(q_{t+1,i} - 3q_{ti}) \log\left(\frac{1}{p_{ti}}\right) &= \sum_{i \in \mathfrak{Q}_t}(q_{t+1,i} - 3q_{ti})\log\left(\frac{1}{p_{ti}}\right) + \sum_{i \notin \mathfrak{Q}_t}(q_{t+1,i} - 3q_{ti})\log\left(\frac{1}{p_{ti}}\right) \\
&\leq \frac{\beta_{t+1}}{\beta_t}\sum_{i \in \mathfrak{Q}_t}(q_{t+1,i} - 3q_{ti})\log\left(\frac{1}{p_{t+1,i}}\right) + 0 \\
&\leq \frac{10}{9}\sum_{i \in \mathfrak{Q}_t}(q_{t+1,i} - 3q_{ti})\log\left(\frac{1}{p_{t+1,i}}\right) \\
&\leq \frac{10}{9}\sum_{i \in \mathfrak{Q}_t}\left(q_{t+1,i} - q_{t+1,i}^{\beta_{t+1}/\beta_t}\right)\log\left(\frac{1}{p_{t+1,i}}\right) \\
&= \frac{10}{9}\sum_{i \in \mathfrak{Q}_t} q_{t+1,i}\left(1 - q_{t+1,i}^{\frac{\beta_{t+1}}{\beta_t}-1}\right)\log\left(\frac{1}{p_{t+1,i}}\right),
\end{aligned}
\tag{34}
$$

where the first inequality follows by $p_{t+1,i} \leq 6p_t^{\beta_t/\beta_{t+1}}$, the second follows by Lemma 6, and the last inequality follows by $q_{t+1,i} \leq 3q_{ti}^{\beta_t/\beta_{t+1}}$. Since for any $\varepsilon > 0$, $x \in [0, 1]$, and $\gamma \in [0, 1]$, it holds that

$$
\begin{aligned}
x(1 - x^\varepsilon) \leq x\log\left(\frac{1}{x^\varepsilon}\right) &= \varepsilon x\log\left(\frac{1}{x}\right) \\
&\leq \varepsilon\left(\left(\log\frac{1}{\gamma} - 1\right)(x - r) + \gamma\log\frac{1}{\gamma}\right) \leq \varepsilon\log\left(\frac{1}{\gamma}\right)(\gamma + (1 - \gamma)x),
\end{aligned}
\tag{35}
$$

setting $\gamma = k/T$ in (35) implies that the RHS of (34) is further bounded as

$$
\begin{aligned}
h_{t+1} - 3h_t \\
\leq \frac{10}{9}\sum_{i \in \mathfrak{Q}_t}\left(\frac{\beta_{t+1}}{\beta_t} - 1\right)\log(T/k)\left(\frac{k}{T} + \left(1 - \frac{k}{T}\right)q_{t+1,i}\right)\log\left(\frac{1}{p_{t+1,i}}\right) \\
\leq \frac{10k}{9}\left(\frac{\beta_{t+1}}{\beta_t} - 1\right)\log(T/k)\sum_{i=1}^{k}\left(\frac{1}{T} + \left(1 - \frac{k}{T}\right)q_{t+1,i}\right)\log\left(\frac{1}{p_{t+1,i}}\right) \\
\leq \frac{20k}{9}\left(\frac{\beta_{t+1}}{\beta_t} - 1\right)\log(T/k)h_{t+1},
\end{aligned}
$$

where the second inequality follows by Lemma 6 and the last inequality follows by the definition of $h_{t+1}$. □

Finally we are ready to prove Theorem 4.

*Proof of Theorem 4.* Fix $i^* \in [k]$ and define $p^* \in \mathcal{P}_k$ by

$$
p^* = \left(1 - \frac{k}{T}\right)e_{i^*} + \frac{1}{T}\mathbf{1}.
$$

Then, using the definition of the algorithm,

$$
\begin{aligned}
\mathrm{Reg}_T(i^*) = \mathbb{E}\left[\sum_{t=1}^{T} \ell_{tA_t} - \sum_{t=1}^{T} \ell_{ti^*}\right] &= \mathbb{E}\left[\sum_{t=1}^{T} \langle \ell_t, p_t - e_{i^*}\rangle\right] \\
&= \mathbb{E}\left[\sum_{t=1}^{T} \langle \ell_t, q_t - e_{i^*}\rangle\right] + \mathbb{E}\left[\gamma \sum_{t=1}^{T} \left\langle \ell_t, \frac{1}{k}\mathbf{1} - q_t\right\rangle\right] \\
&\leq \mathbb{E}\left[\sum_{t=1}^{T} \langle \ell_t, q_t - p^*\rangle\right] + \mathbb{E}\left[\sum_{t=1}^{T} \langle \ell_t, p^* - e_{i^*}\rangle\right] + \gamma T \\
&\leq \mathbb{E}\left[\sum_{t=1}^{T} \langle \widehat{y}_t, q_t - p^*\rangle\right] + 2k\,,
\end{aligned}
$$

where the first inequality follows since $p_t = (1-\gamma)q_t + \frac{\gamma}{k}\mathbf{1}$ and the last inequality follows by the definition of $p^*$ and $\gamma = \frac{k}{T}$. By the standard analysis of the FTRL described in Section 3,

$$
\sum_{t=1}^{T} \langle \widehat{y}_t, q_t - p^*\rangle \leq \sum_{t=1}^{T} \Big(\Phi_t(q_{t+1}) - \Phi_{t+1}(q_{t+1})\Big) + \Phi_{t+1}(p^*) - \Phi_1(q_1)
$$
$$
+ \sum_{t=1}^{T} \Big(\langle q_t - q_{t+1}, \widehat{y}_t\rangle - D_{\Phi_t}(q_{t+1}, q_t)\Big)\,.
$$

We first consider the penalty term. Since $\Phi_t = \frac{1}{\eta_t}\psi^{\mathsf{nS}} + 4\psi^{\mathsf{LB}}$,

$$
\sum_{t=1}^{T} \Big(\Phi_t(q_{t+1}) - \Phi_{t+1}(q_{t+1})\Big) + \Phi_{t+1}(p^*) - \Phi_1(q_1)
$$
$$
\leq \sum_{t=1}^{T} \left(\frac{1}{\eta_{t+1}} - \frac{1}{\eta_t}\right) H(q_{t+1}) + \frac{H(q_1)}{\eta_1} + 4\sum_{i=1}^{k} \log\left(\frac{1}{p_i^*}\right)
$$
$$
\leq \sum_{t=1}^{T} \left(\frac{1}{\eta_{t+1}} - \frac{1}{\eta_t}\right) H(q_{t+1}) + \frac{\log k}{\eta_1} + 4k\log T\,,
$$

where in the last inequality we used the fact that $p_i^* \geq 1/T$ for all $i \in [k]$.

For the stability term, by Lemma 3 with $\delta = 2$,

$$
\sum_{t=1}^{T} \Big(\langle q_t - q_{t+1}, \widehat{y}_t\rangle - D_{\Phi_t}(q_{t+1}, q_t)\Big) \leq 2\sum_{t=1}^{T} \eta_t \nu_t\,.
$$

We will confirm that the assumptions for Part (I) of Theorem 1 are indeed satisfied. By the definition of the learning rate in (8) and $\nu_t \leq \beta_t/2$,

$$
\frac{\sqrt{c_2}}{c_1}(\beta_1 + \beta_t) \geq 9(\beta_1 + \nu_t) \geq \beta_1 + \nu_t\,.
$$

Hence stability condition (S1) of Theorem 1 is satisfied and one can also see that the other assumptions are trivially satisfied. Hence, by Part (I) of Theorem 1,

$$
\begin{aligned}
\sum_{t=1}^{T} \left(\frac{1}{\eta_{t+1}} - \frac{1}{\eta_t}\right) + 2\sum_{t=1}^{T} \eta_t \nu_t &\leq 2\left(c_1 + \frac{2}{c_1}\log\left(1 + \sum_{s=1}^{T} \frac{\nu_s}{\beta_1}\right)\right)\sqrt{c_2 + \sum_{t=1}^{T+1} \nu_t h_{t+1}} \\
&\leq 2\left(c_1 + \frac{2}{c_1}\log\left(1 + \frac{T^2}{\beta_1}\right)\right)\sqrt{c_2 + \sum_{t=1}^{T+1} \nu_t h_{t+1}}\,, \\
&= 2\sqrt{2\log\left(1 + \frac{T^2}{\beta_1}\right)}\sqrt{c_2 + \sum_{t=1}^{T+1} \nu_t h_{t+1}}\,,
\end{aligned}
$$

where in the last inequality we used $\nu_t \leq T$ and in the equality we set $c_1 = \sqrt{2 \log \left(1 + \frac{T^2}{\beta_1}\right)}$.

By summing up the above arguments and Jensen's inequality, we have

$$\text{Reg}_T(i^*) \leq \mathbb{E}\left[2\sqrt{2 \log \left(1 + \frac{T^2}{\beta_1}\right)}\sqrt{c_2 + \sum_{t=1}^{T+1} \nu_t h_{t+1}}\right] + 2k + 4k \log T + \frac{\log k}{\eta_1}$$

$$\leq 2\sqrt{2 \log \left(1 + \frac{T^2}{\beta_1}\right)}\sqrt{c_2 + \mathbb{E}\left[\sum_{t=1}^{T+1} \nu_t h_{t+1}\right]} + 2k + 4k \log T + 15k \log k$$

$$\leq 2\sqrt{2 \log \left(1 + \frac{T^2}{\beta_1}\right)}\sqrt{\mathbb{E}\left[\sum_{t=1}^{T} \nu_t h_{t+1}\right]} + O(k \log T). \tag{36}$$

**Adversarial regime** We first consider the adversarial regime. Recall that $\mathbb{E}\left[\sqrt{\sum_{t=1}^{T} \nu_t}\right] \leq \mathbb{E}\left[\sqrt{\sum_{t=1}^{T} \omega_t}\right] \leq \sqrt{L_2}$ as was done in the proof of Corollary 3. Hence (36) with $h_t \leq 2 \log k$ (since $T \geq 2k$) yields that

$$\text{Reg}_T \leq 4\sqrt{L_2 \log(k) \log \left(1 + \frac{T^2}{\beta_1}\right)} + O(k \log T).$$

**Adversarial regime with a self-bounding constraint** Next we consider the adversarial regime with a self-bounding constraint. We will bound a component of (36). By Lemma 7, $\sqrt{\mathbb{E}\left[\sum_{t=1}^{T} \nu_t h_{t+1}\right]}$ is bounded as

$$X_t := \sqrt{\mathbb{E}\left[\sum_{t=1}^{T} \nu_t h_{t+1}\right]}$$

$$\leq \sqrt{3\,\mathbb{E}\left[\sum_{t=1}^{T} \nu_t h_t\right] + \frac{20k}{9} \log \left(\frac{T}{k}\right)\mathbb{E}\left[\sum_{t=1}^{T} \nu_t \left(\frac{\beta_{t+1}}{\beta_t} - 1\right) h_{t+1}\right]}$$

$$\leq \sqrt{3\,\mathbb{E}\left[\sum_{t=1}^{T} \nu_t h_t\right] + \frac{10k}{9} \log \left(\frac{T}{k}\right)\mathbb{E}\left[\sum_{t=1}^{T} (\beta_{t+1} - \beta_t) h_{t+1}\right]}$$

$$= \sqrt{3\,\mathbb{E}\left[\sum_{t=1}^{T} \nu_t h_t\right] + \frac{10k}{9} \log \left(\frac{T}{k}\right)\mathbb{E}\left[\sum_{t=1}^{T} \frac{c_1 \nu_t h_{t+1}}{\sqrt{c_2 + \sum_{t=1}^{T} \nu_s h_{s+1}}}\right]}$$

$$\leq \sqrt{3\,\mathbb{E}\left[\sum_{t=1}^{T} \nu_t h_t\right] + \frac{20k}{9} \log \left(\frac{T}{k}\right)\mathbb{E}\left[\sqrt{\sum_{t=1}^{T} \nu_t h_{t+1}}\right]}$$

$$= \sqrt{3\,\mathbb{E}\left[\sum_{t=1}^{T} \nu_t h_t\right] + \frac{20k}{9} \log \left(\frac{T}{k}\right) X_t},$$

where the first inequality follows by Lemma 7, the second inequality follows by $\nu_t \leq \beta_t/2$, the last inequality follows by Lemma 8. Since $x \leq \sqrt{a + bx}$ for $x > 0$ implies $x \leq 2\sqrt{a} + b$,

$$X_t \leq 2\sqrt{3\,\mathbb{E}\left[\sum_{t=1}^{T} \nu_t h_t\right]} + \frac{20k}{9} \log \left(\frac{T}{k}\right) = 2\sqrt{3\,\mathbb{E}\left[\sum_{t=1}^{T} \mathbb{E}[\nu_t \,|\, p_t] h_t\right]} + \frac{20k}{9} \log \left(\frac{T}{k}\right)$$

---

**Algorithm 2:** BOBW algorithm with a sparsity-dependent bound in Section 5.2

---
**1** **for** $t = 1, 2, \ldots, T$ **do**
**2** | Compute $q_t$ using (1).
**3** | Sample $A_t \sim p_t$, observe $\ell_{tA_t} \in [-1, 1]$, and compute $\widehat{y}_t$.
**4** | Update $\beta_t$ using (8) based on the bisection method (Algorithm 3).

---

---

**Algorithm 3:** Bisection method for computing $\beta_{t+1}$

---
**1** **input:** $F_t$
**2** left $\leftarrow \beta_t$, right $\leftarrow \beta_t + T$
**3** **while** *true* **do**
**4** | center $\leftarrow$ (left + right)/2
**5** | **if** $F_t(\text{center}) < 0$ **then**
**6** | | left $\leftarrow$ center
**7** | **else if** $F_t(\text{center}) > 0$ **then**
**8** | | right $\leftarrow$ center
**9** | **else**
**10** | | break
**11** **return** center

---

$$\leq 2\sqrt{6s\,\mathbb{E}\left[\sum_{t=1}^{T} H(p_t)\right] + \frac{20k}{9}\log\left(\frac{T}{k}\right)} \;, \tag{37}$$

where we used $\mathbb{E}[\nu_t \mid h_t] \leq \mathbb{E}[\sum_{i=1}^{k} p_{ti}(\ell_{ti}^2/p_{ti})] = \mathbb{E}[\sum_{i \in [k]: \ell_{ti} \neq 0} p_{ti}(\ell_{ti}^2/p_{ti})] \leq s$.

We consider the case of $P(a^*) \geq e$, since otherwise Lemma 2 implies $\sum_{t=1}^{T} H(p_t) \leq e\log(kT)$ and thus the desired bound is trivially obtained. When $P(a^*) \geq e$, Lemma 2 implies that $\sum_{t=1}^{T} H(p_t) \leq P(a^*)\log(kT)$. Then from the self-bounding technique, for any $\lambda \in (0, 1]$ it holds that

$$\begin{aligned}
\mathsf{Reg}_T &= (1 + \lambda)\mathsf{Reg}_T - \lambda\mathsf{Reg}_T \\
&\leq \mathbb{E}\left[(1 + \lambda)O\left(\sqrt{s\log(T)\log(kT)P(a^*)}\right) - \lambda\Delta_{\min}P(a^*)\right] + \lambda C + O(k\log T) \\
&\leq O\left(\frac{(1 + \lambda)^2 s\log(T)\log(kT)}{\lambda\Delta_{\min}} + \lambda C\right) \\
&= O\left(\frac{s\log(T)\log(kT)}{\Delta_{\min}} + \lambda\left(\frac{s\log(T)\log(kT)}{\Delta_{\min}} + C\right) + \frac{1}{\lambda}\frac{s\log(T)\log(kT)}{\Delta_{\min}}\right),
\end{aligned}$$

where the first inequality follows by Lemma 1 and the second inequality follows from $a\sqrt{x} - bx/2 \leq a^2/(2b)$ for $a, b, x \geq 0$. Setting $\lambda \in (0, 1]$ to

$$\lambda = \sqrt{\frac{s\log(T)\log(kT)}{\Delta_{\min}} \Big/ \left(\frac{s\log(T)\log(kT)}{\Delta_{\min}} + C\right)}$$

gives the desired regret bound for the adversarial regime with a self-bounding constraint.

**Stochastic regime** Using $\mathbb{E}[\nu_t \mid h_t] \leq \mathbb{E}[\omega_t \mid h_t] \leq \mathbb{E}[\sum_{i=1}^{k} p_{ti}(\ell_{ti}^2/p_{ti})] = \mathbb{E}[\sum_{i=1}^{k} \ell_{ti}^2]$ in the second inequality of (37) and following the same arguments as the analysis for the adversarial regime with a self-bounding constraint, one can obtain the regret bound for the stochastic regime. $\qquad\square$

### H.3 Discussion on bisection method for computing $\beta_{t+1}$

This section describes the bisection method to compute $\beta_{t+1}$ described in Section 5.2. Recall that $F_t : [\beta_t, \beta_t + T] \to \mathbb{R}$ is defined by the difference of the both sides of the update rule of $(\beta_t)$ in (8):

$$F_t(\alpha) = \alpha - \left( \beta_t + \frac{c_1 \nu_t}{\sqrt{c_2 + \nu_t h_{t+1}(\alpha) + \sum_{s=1}^{t-1} \nu_s h_{s+1}}} \right), \tag{38}$$

where $h_{t+1}(\alpha) = \frac{1}{1-\frac{k}{T}} H(p_{t+1}(\alpha))$, and $p_{t+1}(\alpha)$ is the FTRL output with the regularizer $\Phi_t = \alpha \psi^{\mathsf{nS}} + 4\psi^{\mathsf{LB}}$. Note that $c_1 \nu_t / \sqrt{c_2 + \nu_t h_{t+1}(\alpha) + \sum_{s=1}^{t-1} \nu_s h_{s+1}} \leq c_1 \nu_t / c_2 \leq T/9$ since $\nu_t \leq T$.

Assume that $F_t$ is continuous. Then we can see that there exists $\alpha \in [\beta_t, \beta_t + T]$ such that $F_t(\alpha) = 0$. In fact, if $p_{t A_t} = 0$ then $\beta_{t+1} = \beta_t$, and otherwise, we have $F_t(\beta_t) \leq 0$ and $F_t(\beta_t + T) > 0$. Using the intermediate value theorem with the assumption that $F_t$ is continuous, there indeed exists $\alpha \in [\beta_t, \beta_t + T]$ satisfying $F_t(\alpha) = 0$. We can compute such $\alpha$ by the bisection method. In particular, we first set the range of $\alpha$ to $[\beta_t, \beta_t + T]$, and then iteratively halve it by evaluating the value of $F_t$ at the middle point. Such a bisection method (binary search) is also used in [50], although the computed target is different. The whole BOBW algorithm with the sparsity-dependent bound in Section 5.2 is given in Algorithm 2, and the concrete procedure of the bisection given in Algorithm 3.

Now, all that remains is to show that $F_t$ is continuous. To prove this, it suffices to prove that $h_{t+1}(\alpha) = \frac{1}{1-\frac{k}{T}} H(p_{t+1}(\alpha))$ is continuous with respect to $\alpha$.

**Proposition 2.** $F_t$ in (38) is continuous with respect to $\alpha$.

*Proof of Proposition 2.* Take any $\alpha \in [\beta_t, \beta_t + T]$ and then consider the following optimization problem:

$$q_{t+1}(\alpha) = \underset{q \in \mathcal{P}_k}{\arg\min} \left\langle \sum_{s=1}^{t} \widehat{y}_s, q \right\rangle + \Phi_{t+1}(q),$$

where $\Phi_{t+1} = \alpha \psi^{\mathsf{nS}} + 4\psi^{\mathsf{LB}}$. Now using Corollary 8.1 of Hogan [20] with the fact that the solution of the above optimization problem is unique, $q_{t+1}(\alpha)$ is continuous with respect to $\alpha$. This completes the proof since $p_{t+1}$ is continuous with respect to $q_{t+1}$, $1/T \leq p_{t+1,i}(\alpha) \leq 1 - k/T$, and $H(p)$ is continuous in a neighborhood of $p = p_{t+1}(\alpha)$. $\qquad\square$

## I  Proof of Corollary 6

This section proves Corollary 6, which is the extended result of Corollary 5. Recall that $B = 1/2$ for FI, $B = k/2$ for MAB, and $B = 2mk^2$ for PM-local, and $r_{\mathcal{M}}$ is 1 if $\mathcal{M}$ is FI or MAB, and $2k$ if $\mathcal{M}$ is PM-local, which are appeared in Appendix B. Let $\mathsf{Reg}_T(a) = \mathbb{E}\left[ \sum_{t=1}^{T} \left( \mathcal{L}_{A_t x_t} - \mathcal{L}_{a x_t} \right) \right] = \mathbb{E}\left[ \sum_{t=1}^{T} \langle \ell_{A_t} - \ell_a, e_{x_t} \rangle \right]$ for $a \in [k]$.

*Proof.* Fix $i^* \in [k]$. From Lemma 7 in Tsuchiya et al. [46], if $\eta_t > 0$, we have

$$\mathsf{Reg}_T(i^*) \leq \mathbb{E}\left[ \sum_{t=1}^{T} \left( \frac{1}{\eta_{t+1}} - \frac{1}{\eta_t} \right) H(q_{t+1}) + \frac{H(q_1)}{\eta_1} + \sum_{t=1}^{T} \eta_t V_t' \right]. \tag{39}$$

We will confirm that the assumptions for Part (I) of Theorem 1 are indeed satisfied. Since

$$\frac{\sqrt{c_2 + \bar{z}_t h_1}}{c_1} (\beta_t + \beta_1) \geq \sqrt{\frac{2\bar{V} \log k}{\log(1+T)}} \cdot 2B \sqrt{\frac{\log(1+T)}{\log k}} \geq \sqrt{2} \left( \bar{V} + \bar{V}_t \right),$$

stability condition (S1) is satisfied. One can also see that the other conditions are trivially satisfied. Hence, using Part (I) of Theorem 1, we can bound the RHS of (39) as

$$\mathsf{Reg}_T(i^*) \le \mathbb{E}\left[\left(2c_1 + \frac{1}{c_1}\log\left(1 + \sum_{u=1}^{T}\frac{V_u'}{\bar{V}}\right)\right)\sqrt{\bar{V}H(q_1) + \sum_{t=1}^{T}V_t'H(q_{t+1})}\right] + \frac{H(q_1)}{\eta_1}$$

$$\le \left(2c_1 + \frac{1}{c_1}\log\left(1 + T\right)\right)\sqrt{\mathbb{E}\left[\sum_{t=1}^{T}V_t'H(q_{t+1})\right]} + O\left(\sqrt{\bar{V}\log(k)\log(T)} + B\sqrt{\log(k)\log T}\right)$$

$$= \sqrt{2\log(1+T)}\sqrt{\mathbb{E}\left[\sum_{t=1}^{T}V_t'H(q_{t+1})\right]} + O\left(B\sqrt{\log(k)\log(T)}\right), \tag{40}$$

where the second inequality follows from $V_u'/\bar{V} \le 1$ and in the equality we set $c_1 = \sqrt{\frac{\log(1+T)}{2}}$ and used $\sqrt{\bar{V}} \le B$.

**Adversarial regime**   For the adversarial regime, since $H(q_t) \le \log k$, (40) immediately implies

$$\mathsf{Reg}_T \le \mathbb{E}\left[\sqrt{2\sum_{t=1}^{T}V_t'\log(k)\log(1+T)} + O\left(B\sqrt{\log(k)\log(T)}\right)\right],$$

which is the desired bound.

**Adversarial regime with a self-bounding constraint**   Next, we consider the adversarial regime with a self-bounding constraint. We consider the case of $Q(a^*) \ge \mathrm{e}$, since otherwise Lemma 2 implies $\sum_{t=1}^{T}H(p_t) \le \mathrm{e}\log(kT)$ and thus the desired bound is trivially obtained. When $Q(a^*) \ge \mathrm{e}$, Lemma 2 implies that $\sum_{t=1}^{T}H(q_t) \le Q(a^*)\log(kT)$. Then from the self-bounding technique, for any $\lambda \in (0, 1]$

$$\mathsf{Reg}_T = (1+\lambda)\mathsf{Reg}_T - \lambda\mathsf{Reg}_T$$

$$\le \mathbb{E}\left[(1+\lambda)O\left(\sqrt{\bar{V}\log(T)\log(kT)Q(a^*)}\right) - \frac{\lambda\Delta_{\min}Q(a^*)}{r_{\mathcal{M}}}\right] + \lambda C$$

$$\le (1+\lambda)O\left(\sqrt{\bar{V}\log(T)\log(kT)\bar{Q}(a^*)}\right) - \frac{\lambda\Delta_{\min}\bar{Q}(a^*)}{r_{\mathcal{M}}} + \lambda C$$

$$\le O\left(\frac{(1+\lambda)^2 r_{\mathcal{M}}\log(T)\log(kT)}{\lambda\Delta_{\min}} + \lambda C\right)$$

$$= O\left(\frac{r_{\mathcal{M}}\bar{V}\log(T)\log(kT)}{\Delta_{\min}} + \lambda\left(\frac{r_{\mathcal{M}}\bar{V}\log(T)\log(kT)}{\Delta_{\min}} + C\right) + \frac{1}{\lambda}\frac{r_{\mathcal{M}}\bar{V}\log(T)\log(kT)}{\Delta_{\min}}\right),$$

where the first inequality follows by (40) and Lemma 1 with $c' = r_{\mathcal{M}}$ and the second inequality follows from $a\sqrt{x} - bx/2 \le a^2/(2b)$ for $a, b, x \ge 0$. Setting $\lambda \in (0, 1]$ to

$$\lambda = \sqrt{\frac{r_{\mathcal{M}}\bar{V}\log(T)\log(kT)}{\Delta_{\min}} \Big/ \left(\frac{r_{\mathcal{M}}\bar{V}\log(T)\log(kT)}{\Delta_{\min}} + C\right)}$$

gives the desired bound for the adversarial regime with a self-bounding constraint.   $\square$

## J   Basic lemma

**Lemma 8** ([38, Lemma 4.13]).  *Let $a_0 \ge 0$, $(a_t)_{t=1}^{T}$ be non-negative reals and $f : \mathbb{R}_+ \to \mathbb{R}_+$ be a non-increasing function. Then,*

$$\sum_{t=1}^{T}a_t f\left(a_0 + \sum_{s=1}^{t}a_s\right) \le \int_{a_0}^{\sum_{t=0}^{T}a_t}f(x)\mathrm{d}x.$$

We include the proof for the completeness.

*Proof.* Let $A_t = \sum_{s=0}^{t} a_s$. Then summing the following inequality over $t$ completes the proof:

$$a_t f \left( a_0 + \sum_{s=1}^{t} a_s \right) = a_t f(A_t) = \int_{A_{t-1}}^{A_t} f(A_t) \mathrm{d}x \leq \int_{A_{t-1}}^{A_t} f(x) \mathrm{d}x \,.$$

$\square$

## K  Comparison of the sparsity-dependent bound and the first-order bound for negative losses

If $\ell_t \in [0,1]^k$, the first-order bound by Wei and Luo [49] implies sparsity bounds. This, however, does not hold when $\ell_t \in [-1,1]^k$. In fact, let us consider the case where $\ell_t$ is a zero vector except that only one arm's loss is $-1$ for some $t \in [T]$. Then the sparsity-dependent bound becomes $O(\sqrt{T})$. On the other hand, the first-order bound in [49] is not directly applicable, and we need to transform losses to range $[0,1]$. This implies that the first-order bound becomes $O(\sqrt{kT})$, which is worse than the sparsity-dependent bound.

