# OpenReview forum: "Stability-penalty-adaptive follow-the-regularized-leader: Sparsity, game-dependency, and best-of-both-worlds"
_NeurIPS.cc/2023/Conference — NeurIPS 2023 poster_

### Official Review · Reviewer_Utqx · 2023-07-05

**Soundness:** 4 excellent
**Presentation:** 3 good
**Contribution:** 3 good
**Rating:** 7
**Confidence:** 3

**Summary:**

The paper studies the problem of designing no-regret algorithm with three types of adaptivity to the setting: sparsity, game-dependency, and Best-of-Both-Worlds. To do so, the authors propose a new adaptive learning rate, called Stability-Penalty-Adaptive learning rate for Follow the regularized leader. They provide the first $s$-agnostic regret bound in the adversarial regime, where $s$ is the sparsity of the rewards. Moreover, they design the first Best-of-Both-Worlds algorithm with a sparsity-dependent regret bound. Finally, they extend the analysis to partial monitoring.


**Strengths:**

The paper provides new improved regret bounds for different settings. The paper provides a clear comparison with previous work. The proofs of the theoretical results are not straightforward.

**Weaknesses:**

The paper presents many different results and hence it is hard to follow the paper in some points.

The analysis of the partial monitoring setting is relegated to the last half page. This makes impossible to have an insight on the results from the main paper.

**Questions:**

.

**Limitations:**

yes

---

> ### Author Rebuttal · Authors · 2023-08-09
>
> Thank you for your valuable time and for providing valuable suggestions.
>
> > The paper presents many different results and hence it is hard to follow the paper in some points.
>
> Indeed, multi-armed bandits and partial monitoring are very different, and the results obtained with each problem setting are largely different.
> However, we decided to combine them in one paper because both of their improvements are based on the stability-penalty-adaptive learning rate.
>
> > The analysis of the partial monitoring setting is relegated to the last half page. This makes impossible to have an insight on the results from the main paper.
>
> Due to the space constraint, only the main result (Corollary 3) could be included in the main text for the contribution in partial monitoring.
> Nevertheless, we believe that the differences between the main result and existing results are sufficiently described in the main text through the Section 1, Table 1, and Section 6.
> Furthermore, Section 6 is described as an example to illustrate the wide applicability of the stability-penalty-adaptive learning rate, and the authors believe that the contributions up to Section 5 alone are considered to be sufficiently nontrivial.

---

> > ### Comment · Reviewer_Utqx · 2023-08-20
> >
> > Thank you for the response. My (positive) rating remains unchanged.

---

### Official Review · Reviewer_1eun · 2023-07-09

**Soundness:** 3 good
**Presentation:** 2 fair
**Contribution:** 2 fair
**Rating:** 7
**Confidence:** 3

**Summary:**

This paper proposes a new scheme called SPA to configure adaptive learning rates for FTRL in MAB. Developed from a stability-&penalty-dependent generic regret bound for FTRL, SPA is designed to select learning rates to optimize the bound. SPA is then applied to three scenarios with different types of adaptivity, including sparsity, game-dependency, and BOBW. In particular, SPA eschews the need of knowing the sparsity level in advance as in previous methods, and is shown to achieve a near-optimal regret bound in both stochastic and adversarial settings. Finally, the proposed method is applied to partial monitoring, which achieves a game-dependent bound and BOBW property simultaneously.

**Strengths:**

1. Adaptivity of FTRL is an interesting and important problem. The proposed adaptive methods may improve the applicability of FTRL in real-world scenarios.

2. This paper is technically sound with solid theoretical results.

3. This paper is well written in general and easy to follow.

**Weaknesses:**

1. The design of SPA is rather straightforward, which seems a direct consequence of existing generic regret bound for FTRL. This somehow limits the novelty of the proposed method (though the subsequent extensions to scenarios with various forms of adaptivity are interesting and non-trivial).

**Questions:**

See the "weaknesses".

**Limitations:**

Limitations have been properly discussed.

---

> ### Author Rebuttal · Authors · 2023-08-09
>
> Thank you for your valuable time and for carefully reading our paper.
>
> > The design of SPA is rather straightforward, which seems a direct consequence of existing generic regret bound for FTRL. This somehow limits the novelty of the proposed method (though the subsequent extensions to scenarios with various forms of adaptivity are interesting and non-trivial).
>
> We believe that the idea of a learning rate adaptive to both stability and penalty itself is novel, and its analysis (Theorem 1) involves non-triviality because we need to simultaneously bound the penalty term and stability term.
>
> Furthermore, Theorem 1 and its proof are presented in a simple manner by assuming the stability condition (S1) so that the theorem can be used for both multi-armed bandits and partial monitoring, and in order for the stability condition (S1), it is necessary to make use of time-invariant log-barrier (as described in Remark under Lemma 3), which is non-trivial.
>
> Moreover, Theorem 1 does not immediately lead to Theorem 2 due to the correlation between $\nu_t$ and $h_{t+1}$ as mentioned in "Key elements of the proof" on page 9.
> To address this, we need to analyze the behavior of arm selection probabilities when the regularizer changes (discussed on page 9 and proven in Appendix H), which is highly non-trivial. We will clarify this more in the revision.

---

### Official Review · Reviewer_WVsf · 2023-07-10

**Soundness:** 3 good
**Presentation:** 4 excellent
**Contribution:** 3 good
**Rating:** 6
**Confidence:** 1

**Summary:**

This paper studies how to achieve BOBW regret bounds in sequential decision making problems such multi-armed bandits and partial monitoring games. To this end, this paper comes up with novel SPA learning rates for FTRL. For MAB, this paper shows FTRL with SPA could simultaneously be near-optimal for stochastic and adaptively adversarial sparse MAB, and is agnostic to the sparsity level. For PM, this paper shows it could achieve a regret bound that is both game-dependent and BOBW, and is better than previous results.

**Strengths:**

In addition to the contribution mentioned in summary:
1. The SPA learning rate has an interesting form that may be of interest to the community and might have broader applications.

**Weaknesses:**

1. The improvement upon the previous literature seems not big enough. The main improvement for sparse MAB seems to be removing the sparsity requirement. It is possible I underestimate the hardness of learning sparsity requirement, but since here we only need a multiplicative approximation of the sparsity, may I request the authors to explain why simple ideas such as using successive halving to guess the sparsity level wouldn't work here?

**Questions:**

N/A

---

> ### Author Rebuttal · Authors · 2023-08-09
>
>
> Thank you for your time and for providing a possible alternative procedure for exploiting sparsity in the adversarial regime.
>
>
> > The improvement upon the previous literature seems not big enough. The main improvement for sparse MAB seems to be removing the sparsity requirement. It is possible I underestimate the hardness of learning sparsity requirement, but since here we only need a multiplicative approximation of the sparsity, may I request the authors to explain why simple ideas such as using successive halving to guess the sparsity level wouldn't work here?
>
>
> First of all, the improvement for sparse MAB is not the main contribution of our paper, but only a preparation for Section 5.2, in which we tackle one of our goals to obtain the BOBW guarantee with a sparsity-dependent bound.
> We will emphasize this more in the revised version.
>
> In addition, while improvements in the sparse MAB setting are not a major contribution, we still believe that a successive halving approach would not yield favorable results.
>
> We will assume that successive halving means either
>
> (Approach 1) A procedure that keeps a set containing the sparsity level $s$ and updates its range gradually based on past observations
>
> or
>
> (Approach 2) A procedure based on a method like a hyperparameter tuning that tries several sparsity level $s$ candidates simultaneously and keeps the one with the best result.
>
> We would appreciate it if the reviewer could tell us if these approaches are not what the reviewer expects.
>
> Issues common to both approaches are as follows:
>
> First of all, it is not possible to get a reliable sparsity level $s$ at the halfway point in the adversarial setting, because parameters that are favorable up to the halfway round are not necessarily favorable for the whole round.
>
> Secondly, and importantly, the $L_2$-dependent regret upper bound is stronger than the sparsity level $s$-dependent upper bound.
> In general, $L_2 \leq s T $ holds. Hence, even if we can construct an algorithm that achieves an upper bound of $O(\sqrt{sT \log k} )$ by the above approaches, the regret upper bound of $O(\sqrt{L_2 \log k })$ in our paper is favorable.
> Intuitively, the $L_2$-dependent regret upper bound handles well not only the situation where the loss is sparse in all rounds but also the situation where the loss is sparse when averaged over the entire round.
> To achieve an $L_2$-dependent regret upper bound, we need to cope with the situation where sparsity changes from round to round, which is difficult to do by successive halving.
>
> Finally, it is important to note that even if sparsity can be estimated using the above approaches, it would be significantly difficult to obtain a best-of-both-worlds guarantee simultaneously.
>
> The following is an issue and why such successive halving approaches would be difficult to analyze, although they are not as important as the issues mentioned above.
>
> Issues with Approach 1:
> In the literature, the basic strategy for constructing an algorithm that works adaptively to unknown parameters is to use an adaptive learning rate. On the other hand, doing exploration and exploitation in an adversarial setting while estimating sparsity level $s$ by Approach 1 is likely to make the algorithm and its analysis very complicated.
>
> Issues with Approach 2:
> If based on Approach 2, several bandit algorithms must be run in parallel, but in general, bandit algorithms cannot run in parallel, so it is necessary to split the rounds into blocks, etc.
> Hence the authors expect the design and analysis of such algorithms to become highly non-trivial.

---

> > ### Comment · Reviewer_WVsf · 2023-08-22
> >
> > I have read the rebuttal. The clarification is helpful and I suggest the authors add them to the main text if possible.

---

### Official Review · Reviewer_x1zD · 2023-07-13

**Soundness:** 3 good
**Presentation:** 3 good
**Contribution:** 3 good
**Rating:** 6
**Confidence:** 1

**Summary:**

This paper studies the Follow-the-Regularized-Leader (FTRL) framework for bandit problems. It proposes a learning rate dependent on both stability and penalty terms, which is inspired by the analysis. The proposal is helpful to achieve sparisty, game-dependent and Best of both worlds regret for bandit problems under stochastic and adversarial environments.


**Strengths:**

- The proposed learning rate can be used to design algorithms that achieve sparsity-dependent and BOBW for Multi-arm bandit problem, and game-dependent and BOBW in partial monitoring problem.
- The proposed learning rate might be of independent interest for other online learning problems.
- The preliminaries are well-introduced to average readers.


**Weaknesses:**

- It would be better to provide comparison on regret bounds with $\Delta_{\min}$ and $L_2$ with previous results (some of them don't have these parameters) in Table 1.
- There are too many in-line equations, which might hurt presentation.

**Questions:**

- What would we lose if the learning rate only depends on stability **or** penalty, e.g. the sparsity or gama-denpendent property or BOBW property? It would be better to illustrate as remark for comparison.

**Limitations:**

No limitations are discussed in the paper.

---

> ### Author Rebuttal · Authors · 2023-08-09
>
> Thank you for your time and helpful suggestions.
>
> > It would be better to provide comparison on regret bounds with $\Delta_{\min}$ and $L_2$ with previous results (some of them don't have these parameters) in Table 1.
>
> Thank you for the suggestion for improving Table 1.
> We may not fully understand the reviewer's intent, so if you have time, we would appreciate it if you could be more specific about what perspective you expect the comparison to take.
>
> > There are too many in-line equations, which might hurt presentation.
>
> Due to space limitations, in-line equations are present to some extent.
> However, in-line equations with equation transformations are presented only around Line 327, and the authors believe that the rest of them are considered acceptable.
> In the revised version, we will reduce the number of in-line equations as much as possible.
>
> > What would we lose if the learning rate only depends on stability or penalty, e.g. the sparsity or gama-denpendent property or BOBW property? It would be better to illustrate as remark for comparison.
>
> Thank you for the suggestion.
>
> Section 5 (multi-armed bandits, Theorem 2):
> If the learning rate depends only on stability, then the BOBW guarantee is not obtained,
> and if the learning rate depends only on penalty, then the sparsity-dependent bound is not obtained (i.e., the upper bound of the adversarial regime becomes $O(\sqrt{kT \log k})$ instead of $O(\sqrt{sT \log k})$).
>
> Section 6 (Partial monitoring, Corollary 4):
> If the learning rate depends only on stability, then the BOBW guarantee is not obtained, and if the learning rate depends only on penalty, then the game-dependent bound is not obtained.
>
> In the revised version, we will include this discussion.

---

> > ### Comment · Reviewer_x1zD · 2023-08-14
> >
> > I thank the authors for the response. I have increased the score to reflect the change.
> >
> > - W1: I apologize for not making myself clear. My suggestion is to include discussion to help readers understand results in Table 1. Because parameters $\Delta_{\min}$ does not appear in [9] and $L_2$ does not appear in [25]. We want to understand how the dependency on these parameters of the new result affect the comparison if they behave like a constant or diverge.

---

> > > ### Author Response · Authors · 2023-08-15
> > > **Improving Table 1**
> > >
> > > Thank you for taking the time to clarify the details of your proposal to improve Table 1.
> > > We will include a discussion and comparison of your suggestions in the revised version.
> > > We will provide specific details below.
> > >
> > > > Because parameters $\Delta_{\min}$ does not appear in [9] and $L_2$ does not appear in [25].
> > >
> > > Regarding $\Delta_{\min}$:
> > >
> > > First of all, since both the studies of [9] and [25] consider only the adversarial regime, $\Delta_{\min}$, which can be defined only in the stochastic regime, does not appear in their bounds.
> > > It is well known that the optimal regret for the stochastic regime is approximately $O(k \log (T) / \Delta_{\min})$ (due to limited space, this is described in Line 549 in Appendix).
> > > Note that our upper bound for the stochastic regime in Theorem 2 can be easily improved to $O( s \log (T) \log (kT) / \Delta_{\min})$, as mentioned in the reply to Reviewer TasV.
> > >
> > > Regarding $L_2$:
> > >
> > > Indeed, $L_2$ does not appear in [25], and instead they derive the regret upper bound of $O(\sqrt{sT \log (k/s) })$ with the sparsity level $s$.
> > > However, since $L_2 \leq s T$ holds, a rough comparison with our regret upper bound of $O(\sqrt{L_2 \log k })$ is available.
> > >
> > > The upper bound in [25] is slightly smaller in the worst case, i.e., when $L_2 = s T$, since the inside of the log is $k/s$.
> > > However, in many real-world problems, the regret upper bound of $O(\sqrt{L_2 \log k })$ in our paper is likely to be smaller. This is because the $L_2$-dependent regret upper bound can successfully handle not only the situation where the losses are sparse in all rounds but also the situation where the losses are sparse when averaged over the whole round. Hence, our upper bound can be significantly smaller than the bound in [25].
> > > Furthermore, to improve the inside of the log to $k/s$ instead of $k$, the authors in [25] design a $s$-dependent regularizer based on a priori knowledge of the sparsity level $s$.
> > > On the other hand, our upper bound holds without a priori knowledge of $s$.
> > > Moreover, even if we assume that we know $s$, it is very uncertain whether we can prove one of our main goals, the best-of-both-worlds result (Theorem 2 of our paper), using the regularizer used in [25].
> > >
> > > > We want to understand how the dependency on these parameters of the new result affect the comparison if they behave like a constant or diverge.
> > >
> > > When $\Delta_{\min} \to 0$, although the upper bound in our stochastic regime diverges, the bound for the adversarial regime is still valid even in this case since the stochastic regime is a special case of the adversarial regime.
> > > When $\Delta_{\min}$ is constant, the logarithmic regret upper bound like our bound in Theorem 2 cannot be obtained in [9] and [25].

---

### Official Review · Reviewer_TasV · 2023-07-29

**Soundness:** 3 good
**Presentation:** 2 fair
**Contribution:** 3 good
**Rating:** 6
**Confidence:** 1

**Summary:**

This work considers the Follow-the-regularized-leader paradigm for multi-arm bandit and partial monitoring problems. It proposes a new framework for designing FTRL algorithms with a time-dependent learning rate to generate algorithms that is adaptive to the sparsity of the feedback, to the setting being stochastic/adversarial/adversarial with self-bounding constraints, and to the game difficulty of a PM problem.
Specifically, it proposes to choose the learning rate $\eta_t$ based on the penalty term (changes in the objective function due to the change of the time-dependent regularizer) and the stability term (a measure of progress) observed in previous iterations.

**Strengths:**

The overall idea of this paper has many implications in different settings: for example, in Section 5.1, it generates algorithms with near-optimal guarantees in the setting of adversarial bandits with sparse feedback with/without restrictions on the signs of the feedback; in Section 5.2 it leads to an algorithm that is adaptive to the sparsity and to the adversarial-ness.

**Weaknesses:**

While all the notations used in this paper are defined before being used, it would be great if some context/discussion can be provided before diving into the statements and analyses. For example, what is the intuitive way to see that $opt\prime_{q_t}(\eta_t)$ captures the game difficulty? It would also be nice to explicitly describe how the learning rate is used in the FTRL updates.

**Questions:**

The algorithm analyzed in Section 5.2 achieves BOBW and adaptivity to sparsity in the adversarial setting. Is it possible to also have a notion of sparsity in the stochastic setting?

---

> ### Author Rebuttal · Authors · 2023-08-09
>
> Thank you for your time and insightful suggestions.
>
> > While all the notations used in this paper are defined before being used, it would be great if some context/discussion can be provided before diving into the statements and analyses. For example, what is the intuitive way to see that $\mathrm{opt}'_{q_t}(\eta_t)$ captures the game difficulty?
>
> Thank you for the suggestion.
> In the revised version, we will include more intuitive explanations.
>
> The intuitive interpretation of how $\mathrm{opt}'_{q_t}(\eta_t)$ captures game difficulty is as follows:
>
> In partial monitoring (and many other sequential decision problems), it is necessary to exploit the information currently available, while at the same time constructing a loss estimator that appropriately balances its bias and variance.
> If we can construct such an algorithm, then the algorithm can achieve a better regret.
>
> The first, second, and third terms of $\mathsf{ST}(p,G)$, the objective function that determines $\mathrm{opt}'(\eta_t)$,
> correspond to the degree of the exploitation, bias, and variance terms, respectively.
> Therefore, the smaller $\mathsf{ST}(p,G)$ can become by optimizing $p$ and $G$, the smaller the regret can become, and thus $\mathrm{opt}'_{q_t}(\eta_t)$ captures the game difficulty.
>
> We will include this discussion in the revised version.
>
>
> > It would also be nice to explicitly describe how the learning rate is used in the FTRL updates.
>
> Thank you for the suggestion.
> As defined in Eq. (4), $\eta_t$ is defined as $\eta_t = 1/\beta_t$, and this $\eta_t$ is used to define the regularizer $\Phi_t$ for FTRL.
> In the revised version, we recall these definitions again when defining $\beta_t$ (in Eqs. (6), (7), (8)).
>
> > The algorithm analyzed in Section 5.2 achieves BOBW and adaptivity to sparsity in the adversarial setting. Is it possible to also have a notion of sparsity in the stochastic setting?
>
> Thank you for the insightful suggestion.
> We only considered exploiting sparsity in the adversarial setting and did not in the stochastic setting.
>
> Exploiting the sparsity in the stochastic setting can be accomplished with a very minor modification of the analysis.
> In the equation below Line 766, we use $\mathbb{E}[\nu_t | h_t] \leq k$ without considering the sparsity of losses, but instead we can just use $\mathbb{E}[\nu_t | h_t] \leq s$.
> This indeed holds since
> $ \mathbb{E}[\nu_t | h_t] \leq \sum_{i \in [k]} p_{ti} (\ell_{ti}^2 / p_{ti}) = \sum_{i \in [k] \colon \ell_{ti} \neq 0} p_{ti}  (\ell_{ti}^2 / p_{ti}) \leq s$.
> This allows us to obtain a regret upper bound of $O( s \log (T) \log (kT) / \Delta_{\min})$ instead of $O( k \log (T) \log (kT) / \Delta_{\min})$ in the stochastic setting.
>
> The authors found that Kwon et al. (2017) provides an example of how to utilize sparsity in the stochastic setting of MAB, but they define $s$ as the number of arms with rewards smaller than or equal to 0 (i.e., loss larger than 0), and hence the definition of sparsity is different from $s$ in our paper.
>
> The revised version will include this result with appropriate citations to references such as Kwon et al. (2017).
>
> (Kwon et al. 2017) Sparse Stochastic Bandits, COLT2017

---

> > ### Comment · Reviewer_TasV · 2023-08-19
> >
> > Thank you for the response. The score has been increased. My confidence score remains unchanged as I don't have the expertise to evaluate this paper properly.

---

### Decision · Program_Chairs · 2023-09-21

**Decision:**

Accept (poster)

**Comment:**

In this paper, the authors investigate Follow the-Regularized-Leader (FTRL) for bandits. The primary contribution is the introduction of a Stability-Penalty-Adaptive (SPA) learning rate for FTRL, leading to a regret bound that jointly depends on the stability and penalty of the algorithm. Following this, the authors develop several algorithms characterized by three types of adaptivity: sparsity, game-dependency, and Best-of-Both-Worlds (BOBW).

Specifically, they first establish a regret bound that is dependent on sparsity but without requiring prior knowledge of the sparsity level. Next, they introduce a BOBW algorithm, achieving a near-optimal regret bound in both stochastic and adversarial settings. Lastly, they demonstrate a BOBW guarantee paired with a game-dependent bound for partial monitoring. In summary, this paper offers substantive contributions to the ongoing research in the fields of FTRL and bandits, advancing our understanding in several aspects.